# Diverse Prototypical Ensembles Improve Robustness to Subpopulation Shift

**Minh Nguyen Nhat To** [1 2]  **Paul F R Wilson** [3]  **Viet Nguyen** [4]  **Mohamed Harmanani** [3]  **Michael Cooper** [4]
**Fahimeh Fooladgar** [1]  **Purang Abolmaesumi** [1]  **Parvin Mousavi** [3 2]  **Rahul G. Krishnan** [4 2]

## Abstract

Subpopulation shift, characterized by a disparity in subpopulation distribution between the training and target datasets, can significantly degrade the performance of machine learning models. Current solutions to subpopulation shift involve modifying empirical risk minimization with re-weighting strategies to improve generalization. This strategy relies on assumptions about the number and nature of subpopulations and annotations on group membership, which are unavailable for many real-world datasets. Instead, we propose using an ensemble of diverse classifiers to adaptively capture risk associated with subpopulations. Given a feature extractor network, we replace its standard linear classification layer with a mixture of prototypical classifiers, where each member is trained to classify the data while focusing on different features and samples from other members. In empirical evaluation on nine real-world datasets, covering diverse domains and kinds of subpopulation shift, our method of Diverse Prototypical Ensembles (DPEs) often outperforms the prior state-of-the-art in worst-group accuracy. The code is available at https://github.com/minhto2802/dpe4subpop.

## 1. Introduction

The performance of machine learning models is known to degrade substantially in the presence of distribution shifts between training and deployment (Koh et al., 2021). One common form of distribution shift is *subpopulation shift* (Yang et al., 2023), where the proportions of subgroups vary be-

tween training and target distributions. The study quantitatively categorized subpopulation shifts into four fundamental types: (1) Spurious Correlations – non-causal features mistakenly influence predictions; (2) Attribute Imbalance – certain attribute values appear more frequently than others; (3) Class Imbalance – some labels are significantly underrepresented; and (4) Attribute Generalization – models encounter previously unseen attribute values at test time. These distinct shift types highlight the diverse challenges of subpopulation robustness and motivate approaches aimed at improving worst-group performance.

Naïve training using empirical risk minimization (ERM) can result in classifiers achieving good training loss but not generalizing, performing poorly in challenging or underrepresented subpopulations (Sagawa et al., 2019; Santurkar et al., 2021). Such failures can be catastrophic in performance-critical real-world applications such as medical diagnostics (Oakden-Rayner et al., 2020), autonomous driving (Yu et al., 2020), and insurance risk assessment (Boodhun & Jayabalan, 2018). A commonly cited example is networks that learn to recognize pneumonia relying on hospital-specific meta-tokens on X-ray scans due to their common co-occurrence in training data (Zech et al., 2018), thus struggling to generalize to new data without the tags.

Methods to mitigate subpopulation shift focus on sampling-based (Idrissi et al., 2022; Kirichenko et al., 2022) or loss-based (Michel et al., 2022; Han et al., 2022; Sagawa et al., 2020) re-weighting strategies, such as up-weighting minority subpopulations to encourage the model to learn decision boundaries that adequately classify each subpopulation group. But these often require subgroup annotations (Rudner et al., 2024; Kirichenko et al., 2022), which are rarely available or sufficiently granular in real-world datasets, or explicit identification of minority groups (Liu et al., 2021; Zhang et al., 2022), which significantly increases complexity and training time while struggling to generalize to unseen subgroups (Yang et al., 2023; Zhang et al., 2022).

We propose **D**iversified **P**rototypical **E**nsembles (DPE). Subpopulation shifts results in model degradation because a single classifier typically focuses on the majority classes or subgroups over the minority ones. By turning to ensembles we can capture multiple different decision boundaries. But

---

[1]Department of Electrical and Computer Engineering, University of British Columbia, Vancouver, Canada [2]Vector Institute, Toronto, Canada [3]School of Computing, Queen's University, Kingston, Canada [4]Department of Computer Science, University of Toronto, Toronto, Canada. Correspondence to: Minh Nguyen Nhat To <mtrcl@student.ubc.ca>.

*Proceedings of the 42nd International Conference on Machine Learning*, Vancouver, Canada. PMLR 267, 2025. Copyright 2025 by the author(s).

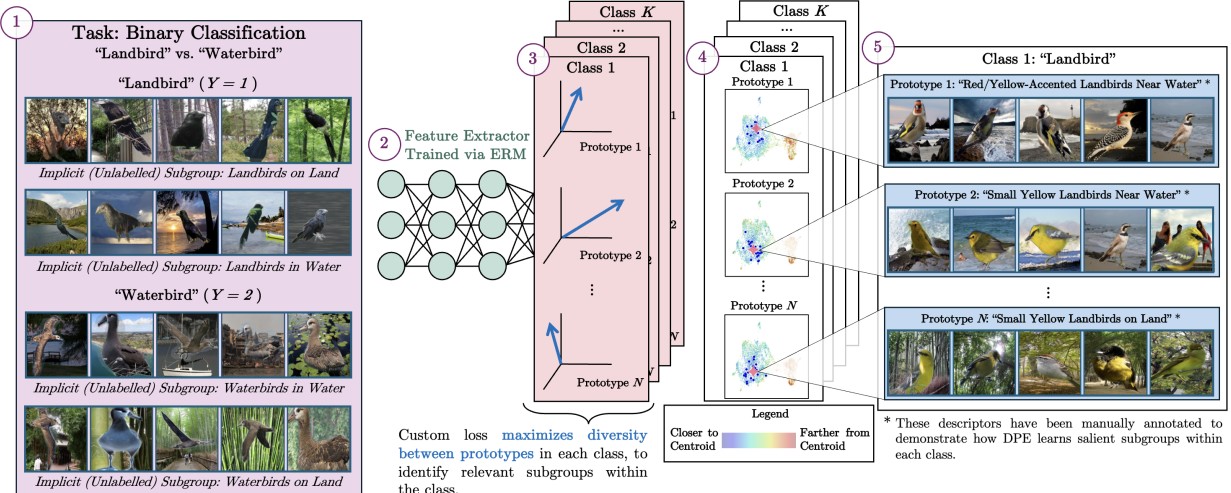

Figure 1. High-level overview of our method. **(1)** Binary classification with implicit (unannotated) subgroups. We aim to natively detect and correct for subpopulation shifts without prior subgroup knowledge. **(2)** Given a **frozen feature extractor,** $f(\cdot)$, we train **(3)** an ensemble of $N$ prototype classifiers for each of the $K$ classes to identify distinct sub-groups. These classifiers are trained using $\mathcal{L}_{\text{IPS}}$ (Equation 4) to maximize prototype diversity, ensuring robust subpopulation capture within each class. **(4)** A low-dimensional projection of the centroids and proximal images for class "Landbird" in *Waterbirds*. The **learned centroids for each ensemble member** reveal unique latent subpopulations. Points **closest to each centroid** appear in blue, while points **farther away** are in red. The **closest few points** are shown in dark blue, with corresponding images visualized in **(5)**. **(5)** Visualization confirms DPE's ability to capture salient subgroups. We have manually annotated the theme associated with each learned prototype centroid. The closest points to each centroid exhibit thematic consistency, aligning with implicit data subgroups (*e.g.*, birds "on land" vs. "in water").

a naive algorithm for learning ensembles may not necessarily encourage each member to capture a different decision boundary. We take inspiration from recent work studying out of distribution detection and improving generalization (Ginsberg et al., 2022; Pagliardini et al., 2022) by encouraging diversity among members of an ensembles. If ERM encourages focusing on the majority class' decision boundary, explicitly encouraging diversity could encourage subsequent members of the ensemble to capture the different decision boundaries corresponding to subgroups *even when* labels are unavailable or the number, identity or distribution of the subpopulations is unknown. Given a feature extractor obtained from standard ERM training, we replace its classification head with an ensemble of prototype classifiers. Each new prototype serves to predict the class of nearby points in the latent space. An inter-prototype similarity loss is used to promote diversity among the ensemble members, resulting in the discovery of many different decision boundaries, each better suited to different subpopulations. This approach yields a model that is robust to subpopulation shifts, as it can adapt to diverse data distributions and maintain classification accuracy across a broad range of subpopulations. Our primary contributions are summarized as follows:

- We propose a novel differentiable **end-to-end** solution to subpopulation shift based on the idea of diversified ensemble, training a collection of diversified predictors

to discover and classify subpopulations in the data. Our method improves robustness to subpopulation shifts, even under unknown subgroup annotations, subpopulation number, identity, or distribution.

- Our solution replaces the linear layer of a trained network with the Diversified Prototypical Ensemble (DPE), a collection of distance-based classifiers that incorporate explicit diversification through a loss term and sampling strategy.

- We empirically validate DPE using nine real-world datasets proposed in (Yang et al., 2023) to assess robustness against different types of subpopulation shifts. Our results show DPE's superior performance over prior state-of-the-art methods, including in challenging cases like attribute generalization and imbalance.

## 2. Related Work

**Subpopulation Shift**   Sugiyama et al. (2008) characterized the effects of the covariate shift on a variety of performance metrics. Rabanser et al. (2019); Li et al. (2021) showed how metrics changed relatively among the subgroups of the data under the covariate shift. Yang et al. (2023) created a unified framework for the analysis of the performance of models under various types of subpopulation shift.

Several methods have been proposed to mitigate the effects of spurious correlations in DL models. By optimizing for the worst-performing subgroup, Group Distributionally Robust Optimization (gDRO) (Sagawa et al., 2019; 2020) forces the model to learn features that are robust predictive across all subgroups, while UnLearning from Experience (ULE) (Mitchell et al., 2024) trains a student model with no constraints to pursue the spurious correlations in the data while a teacher model is trained to solve the same problem while avoiding the student's mistakes. Hierarchical methods tackle the same problem by defining a structured feature space via predefined label taxonomies (Mukherjee et al., 2023) to improve worst-group generalization under structured shift. Liang et al. (2022) propose a prototype-based incremental learning method that sequentially adapts classifiers to new subpopulations using margin-enforce loss, aiming to balance acquisition and forgetting in the presence of subpopulation shift. Just train twice (JTT) (Liu et al., 2021) trains the model twice, with the second stage minimizing the loss over training examples from a resampled dataset that are misclassified at the end of the first stage. Idrissi et al. (2022); Deng et al. (2024) highlight the effects of simple data balancing and subgroup-balanced sampling on worst-group accuracy, showing that simple reweighting and resampling can achieve state-of-the-art performance on most benchmarks. Unlike this prior work, which often relies on explicit subgroup annotations, our method automatically discovers subgroups to accommodate subpopulation shift without prior knowledge of subgroup identities.

**Feature Learning Under Subpopulation Shift**   Geirhos et al. (2020) investigate when and how neural networks learn spurious features that potentially cause performance degradation. Xing et al. (2021); Salazar et al. (2021); Tian et al. (2022) either implicitly or explicitly learn features while optimizing for various fairness metrics. Kirichenko et al. (2022) demonstrated that core information can be extracted from feature representations learned by standard ERM even when spurious correlation exists in training data. Izmailov et al. (2022) explores last-layer retraining by introducing deep feature reweighting (DFR), further highlighting that ERM-learned features are competitive with those from group robustness methods as DFR achieves state-of-the-art results on many vision and NLP benchmarks. Qiu et al. (2023); LaBonte et al. (2024) provide ablation studies of the last-layer retraining paradigm and propose different reweighting schemes to improve group robustness and optimize execution time. DPE builds on these insights by freezing the feature extractor and replacing the ERM-based classifiers with distance-based classifiers to effectively capture subpopulation structures.

**Prototypical Networks and Representations**   Prototype methods use data prototypes as representatives values in their class. Prototypical networks (Snell et al., 2017) approximated a latent space using a neural network, where classification was performed using distances to latent prototypes of each class. The automated discovery of prototype latent spaces has found success in supervised vision (Yang et al., 2018) and image segmentation (Dong & Xing, 2018). Prototype-based classification has proven effective in few-shot learning (Snell et al., 2017) and robustness against shortcut learning (Wei et al., 2023) by leveraging distance-based representations. Their results show promising applications of prototypical classifiers in addressing subpopulation shift, since most successful methods rely on the limited availability of balanced group-annotated data. Our method extends these findings by training multiple prototypes per class, allowing the ensemble to adapt to heterogeneous subpopulations within each category.

**Ensemble Diversity**   Diversification of ensemble members is thought to improve the robustness and generalization of ensemble learning methods (Fort et al., 2019). Several studies have explored methods to promote diversity and extract hidden patterns in the latent space, such as varying training data subsets (bagging and boosting) (Chawla et al., 2003; Seiffert et al., 2009; Zhang et al., 2019), or using regularizers (Xie et al., 2017; Xie, 2015). Diversification through disambiguation (DivDis) (Lee et al., 2022) aims to enhance ensemble diversity by resolving uncertainties that arise from ambiguous data representations. DivDis introduces heterogeneity by encouraging models to specialize in different interpretations of overlapping or uncertain instances. D-BAT (Pagliardini et al., 2023) learns a diverse ensemble of models by pushing the models to agree on the source distribution while disagreeing on out-of-distribution inputs. Gating networks (Riquelme et al., 2021; Zhou et al., 2022) determine which experts in mixture-of-experts frameworks to activate for a given input, thus promoting diversity by encouraging specialization in different regions of the input space. Our approach explicitly enforces this diversity within a prototypical ensemble, ensuring that each classifier captures complementary subpopulation-specific features, leading to improved worst-group accuracy.

## 3. Subpopulation Prototypical Ensemble

**Motivation**   To minimize training loss, a learned decision boundary often exploits features that correlate with the majority subpopulations but fail to generalize. For example, a classifier that distinguishes "cow" from "camel" using background cues such as "grassy" versus "sandy" performs poorly on minority cases, such as cows in desert environments. In contrast, forcing the model to consider multiple, diverse decision rules—each relying on distinct feature subsets—increases the likelihood of discovering a rule that generalizes across subpopulations (e.g., "hump" versus "no

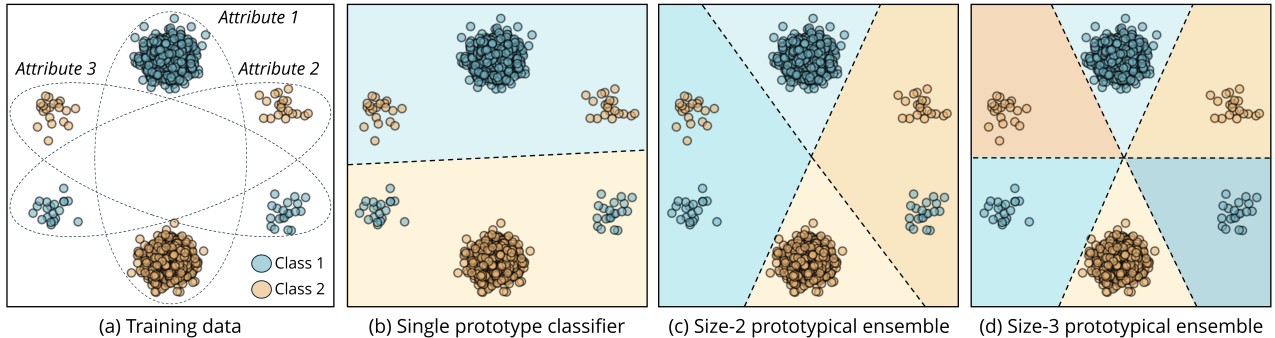

| (a) Training data | (b) Single prototype classifier | (c) Size-2 prototypical ensemble | (d) Size-3 prototypical ensemble |

*Figure 2.* Motivation of DPE. (a) The synthetic training data consists of two classes, with major subgroups containing Attribute 1 and minority subgroups containing Attributes 2 and 3. Training a single model on the entire dataset leads to suboptimal decision boundaries, focusing primarily on the major subgroups; (b, c, d) as the number of models in the prototypical ensemble increases, where each member is trained to classify based on a distinct attribute, decision boundaries become more refined, improving classification across subpopulations.

hump," and "beige" versus "spotty").

This intuition serves as the basis for our *Diversified Prototypical Ensemble* (DPE). Given a set of features, DPE learns an ensemble of distance-based prototypical classifiers, each providing a plausible decision rule while exploiting different features than other members of the ensemble. We argue that even when one decision rule fails on a given subpopulation, other ensemble members are likely to succeed. As a result, the ensemble exhibits improved generalization under subpopulation shift (Fig. 2). We choose prototypical classifiers as base learners due to their ability to preserve feature space geometry in the limited-data regime compared to ERM (Snell et al., 2017). This approach augments contemporary methods of mitigating subpopulation shift that require re-training the classifier using a small, subgroup-annotated subset of the validation set (Yang et al., 2023).

Our method proceeds in two stages. First, we train a general-purpose backbone feature extractor using ERM on the training set. Next, we select a subset of data from the validation set (Izmailov et al., 2022), and train DPE on this subset using the fixed features extracted by the backbone. Our DPE ensemble is characterized by the use of distance-based prototypical classifiers, explicit (loss-based) diversity regularization, and implicit (sampling-based) diversity regularization.

**Two-Stage Training** The first step of our pipeline entails training a feature extractor $f : \mathbb{R}^n \to \mathbb{R}^d$ to map raw inputs into a lower-dimensional representation space. We train our feature extractor to convergence on the full training set, following prior results that classical ERM suffices to produce strong features despite subpopulation shift (Izmailov et al., 2022; Kirichenko et al., 2022).

We then freeze $f$ and and train our Diversified Prototypical Ensemble (DPE) classifier using a small, held-out sub-set of data. In our implementation, following similar approaches in the literature, we use a subset of the validation set (Izmailov et al., 2022; Kirichenko et al., 2022), and for robustness, we use a class-balanced subset of these data. Crucially, unlike many existing methods, our approach *does not require subgroup attribute annotations at this stage*, although if such annotations are available, we recommend using them to construct an attribute-balanced subset for this stage, which functionally serves to combine our more robust DPE (described in the following sections) with classifier retraining methods like (Kang et al., 2020; Kirichenko et al., 2022).

**Prototype Classifier** Given a feature extractor $f : \mathbb{R}^n \to \mathbb{R}^d$ and a collection of $K$ classes, a prototype classifier defines a set of learnable prototypes $\{p^{(i)} : i = 1, ..., K\} \subset \mathbb{R}^d$. For an input $X$, the classifier computes the probability of label $y$ based on the distance between the extracted feature $f(X)$ and each prototype,

$$P(y|X) = \frac{\exp\left(-D(f(X), p^{(y)})\right)}{\sum_{i=1}^{K} \exp\left(-D(f(X), p^{(i)})\right)}, \quad (1)$$

where $D : \mathbb{R}^d \times \mathbb{R}^d \to \mathbb{R}$ is a scaled Euclidean distance between normalized vectors, as in Macêdo & Ludermir (2021),

$$D(x, y) = |d_s| \cdot \left\| \frac{x}{\|x\|} - \frac{y}{\|y\|} \right\|_2,$$

where $d_s$ is a learnable scaling factor. Then, the loss for each data-label pair $(X, y)$ is,

$$\mathcal{L}(X, y) = -\log\left(\frac{\exp\left(-D(f_\theta(X), p^{(y)}/\tau)\right)}{\sum_{i=1}^{K} \exp\left(-D(f_\theta(X), p^{(i)}/\tau)\right)}\right), \quad (2)$$

where $\tau$ is a temperature hyperparameter. We initialize prototypes randomly with mean $\mu = 0$ and standard deviation

$\sigma = 0.01$, and train them by minimizing Equation 2 using stochastic gradient descent.

**Prototypical Ensemble**   Rather than a single prototype per class, our method uses an ensemble of $N$ prototypes per class, yielding a collection $\{p_j^{(i)}\}_{i=1,...,K,\ j=1,...,N}$. This ensemble classifier implements the following classification rule to aggregate predictions from the $N$ prototype-based classifiers:

$$\hat{y} = \arg \max_{k \in \{1,...,K\}} \frac{1}{N} \sum_{j=1}^{N} P_j^{(k)}(y|X), \qquad (3)$$

where $P_j^{(k)}(y|X)$ is the prediction from the $j$th ensemble member under Equation 1.

**Ensemble Diversification**   To encourage diverse decision rules across ensemble members, we employ two prototype diversification strategies: an explicit inter-prototype similarity (IPS) loss and implicit diversification via bootstrap aggregation. Without these, naïvely training each ensemble member may lead to redundant decision boundaries between members of the ensemble.

The IPS loss decorrelates the representations of different prototypes of the same class. For the $n$'th ensemble member, this loss is,

$$\mathcal{L}_{\text{IPS}} = \sum_{k=1}^{K} \sum_{i=1}^{n} \sum_{j=1}^{n} \mathbb{1}_{\{i \neq j\}} \frac{|\langle p_i^{(k)}, p_j^{(k)} \rangle|}{n \cdot d}, \qquad (4)$$

where $\mathbb{1}$ denotes the indicator function, and $\langle \cdot, \cdot \rangle$ denotes the Euclidean inner product. In $\mathcal{L}_{\text{IPS}}$, note that terms are scaled by $n$ and $d$, the number of ensemble members and embedding dimensions, respectively. At each ensemble stage $n$, we simultaneously optimize all prototypes associated with that stage $\{p_n^{(k)}\}_{k=1,...,K}$ while freezing the prototypes that have been optimized in previous stages, $\{p_{n'}^{(k)}\}_{k=1,...,K,\ n'=1,...,n-1}$. We optimize each prototype via stochastic gradient descent on the sum of $\mathcal{L}_{\text{IPS}}$ and $\mathcal{L}(X, y)$ (Equation 2).

In parallel, we apply bootstrap aggregation by training each ensemble member on a different class-balanced subset of the validation data. These random subsets expose each prototype to slightly different distributions, implicitly encouraging diversity in the learned decision boundaries. The ensemble is thus trained **end-to-end** by sequentially solving the joint prototype and IPS loss for each new ensemble member.

## 4. Experiments

### 4.1. Datasets

We conduct comprehensive experiments across nine real-world datasets spanning multiple domains to evaluate the robustness of DPE against subpopulation shift. These datasets represent diverse challenges, including spurious correlations, attribute and class imbalances, and attribute generalization, all common sources of subpopulation shift. The datasets are chosen based on their ability to test various robustness aspects in machine learning models, particularly under settings where attribute information may be unknown or imbalanced. Specifically, the evaluation includes WA-TERBIRDS (Wah et al., 2011), CELEBA (Liu et al., 2015), METASHIFT (Liang & Zou, 2022), IMAGENETBG (Xiao et al., 2021), NICO++ (Zhang et al., 2023), LIVING17 (Santurkar et al., 2021), CHEXPERT (Irvin et al., 2019), CIVIL-COMMENTS (Borkan et al., 2019), and MULTINLI (Schuhmann et al., 2022). We use the same training/validation/test splits given by (Yang et al., 2023). More details of all datasets are provided in Appendix A.1.

### 4.2. Attribute Availability

Attribute availability significantly affects a model's ability to handle subpopulation shift. Following recent works (Kirichenko et al., 2022; Rudner et al., 2024), we consider a scenario where the group-annotated validation set is available and can be used for model selection, hyperparameters tuning, and re-training the classifier head when freezing the feature extractor. Yang et al. (2023) showed that that most methods for robust learning with subpopulation shift greatly benefit from access to a small set of group-annotated data to improve performance on underrepresented groups. However, in many real-world scenarios, attribute annotations are not available during training or validation. In such cases, models must rely on the inherent structure of the data to handle subpopulation shift.

DPE is designed to perform effectively in both settings. We evaluate DPE's capability to identify and adapt to potential subpopulations based on inherent data distribution alone in the absence of explicit subgroup annotations, and its efficacy in utilizing the subgroup annotations to specifically tailor the prototype representations on well-defined subpopulations. Specifically, we use all the aforementioned datasets in experiments *without* subgroup annotations, and only WATERBIRDS, CELEBA, CIVILCOMMENTS, MULTINLI, METASFHIT, and CHEXPERT in experiments *with* subgroup annotations.

### 4.3. Baselines

To rigorously evaluate the effectiveness of DPE in handling subpopulation shift, we compare its performance against

*Table 1.* Worst-group accuracy (WGA) on the test set without subgroup annotations. The top half reproduces baselines from Subpop-Bench (Yang et al., 2023) using the same ERM backbone, while the bottom half includes results from original papers and our method applied to a stronger ERM* backbone. DPE denotes our diversified prototypical ensemble. Best and second-best values are bolded within each half of the table. The best result per group is **underlined and bolded**, the second-best is in **bold**. "-" indicates results not reported.

| Algorithm | WATERBIRDS | CELEBA | CIVILCOMMENTS | MULTINLI | METASHIFT | CHEXPERT | IMAGENETBG | NICO++ | LIVING17 |
|---|---|---|---|---|---|---|---|---|---|
| ERM | 69.1±4.7 | 57.6±0.8 | 63.2±1.2 | **66.4**±2.3 | 82.1±0.8 | 41.7±3.4 | 76.8±0.9 | 35.0±4.1 | 48.0±1.5 |
| CRT | 76.3±0.8 | 69.6±0.7 | **67.8**±0.3 | 65.4±0.2 | 83.1±0.0 | 74.6±0.4 | **78.2**±0.5 | 33.3±0.0 | – |
| ReWeightCRT | 76.3±0.2 | 70.7±0.6 | 64.7±0.2 | 65.2±0.2 | **85.1**±0.4 | 75.1±0.2 | 77.5±0.7 | 33.3±0.0 | – |
| DFR | **89.0**±0.2 | **73.7**±0.8 | 64.4±0.1 | 63.8±0.0 | 81.4±0.1 | **75.8**±0.3 | 74.4±1.8 | **38.0**±3.8 | – |
| ERM + DPE | **91.0**±0.5 | **81.9**±0.2 | **69.9**±0.9 | **69.3**±0.8 | 84.1±1.5 | – | **87.9**±0.6 | **50.0**±0.0 | **54.0**±4.0 |
| ERM* | 77.9±3.0 | 66.5±2.6 | **69.4**±1.2 | 66.5±0.7 | 80.0±0.0 | 75.6±0.4 | 86.4±0.8 | 33.3±0.0 | 53.3±0.9 |
| RWY | 86.1±0.7 | **82.9**±2.2 | 67.5±0.6 | 68.0±1.9 | – | – | – | – | – |
| AFR | **90.4**±1.1 | 82.0±0.5 | 68.7±0.6 | **73.4**±0.6 | – | – | – | – | – |
| ERM* + DPE | **94.1**±0.2 | **84.6**±0.8 | 68.9±0.6 | 70.9±0.8 | **83.6**±0.9 | **76.8**±0.1 | **88.1**±0.7 | **50.0**±0.0 | **63.0**±1.7 |

baselines and current state-of-the-art methods for subpopulation shift robustness, including Empirical Risk Minimization (**ERM**), Classifier Re-train (**CRT**, **ReweightCRT**) (Kang et al., 2020), Deep Feature Reweighting (**DFR**) (Kirichenko et al., 2022), Just Train Twice (**JTT**) (Liu et al., 2021) and Correct-n-Contrast (**CnC**) (Zhang et al., 2022), **RWY** (Idrissi et al., 2022), Automatic Feature Reweighting (**AFR**) (Qiu et al., 2023), and Group-Aware Priors (**GAP**) (Rudner et al., 2024). See more details in Appendix A.3.

## 4.4. Implementation

We prepared each dataset in alignment with established benchmarks on subpopulation shift, as outlined in Yang et al. (2023). We adopted pretrained ResNet-50 for image data and BERT for textual data to facilitate a direct comparison with state-of-the-art benchmarks on subpopulation shift robustness. Hyperparameters related to DPE (e.g., $\tau$ and $\alpha$) are tuned using a held-out subset of the validation set, which is split into training and validation folds for tuning. For consistency and rigor, both hyperparameter tuning and model selection are based on the worst-group accuracy within the validation fold. After hyperparameter selection, we retrain the prototypical ensemble on the full validation set using the selected hyperparameters.

Each two-stage training cycle is repeated three times with three different seeds to ensure the stability of the method. Compared to the initial training stage implemented in Yang et al. (2023) (reported as ERM), our initial training stage involves stronger augmentation (random-crop/resize, horizontal flip) for visual datasets and longer training time for all datasets (similar to Idrissi et al. (2022), reported as ERM*). In the second training stage, we trained 15 prototypes sequentially for all datasets. The full implementation is detailed in Appendix B and our code base.

## 5. Results and Discussion

### 5.1. DPE Achieves Strong Subpopulation Shift Robustness Without Subgroup Annotations

When attributes are unknown during both the training and validation phases, the performance of most methods degrades significantly under subpopulation shift. Table 1 presents the worst-group accuracy (WGA) across multiple datasets, illustrating the challenges that various methods face in adapting to subpopulation shift without subgroup annotations. ERM and ERM* represent naïve baselines of performance in that they implement no mechanism of explicitly accommodating subpopulation shift. Specifically, ERM achieves a WGA of only 69.1% on the WATERBIRDS dataset and 63.2% on CIVILCOMMENTS. Additionally, reweighting approaches like CRT and ReWeightCRT struggle to adapt in these scenarios, as they depend on access to accurate attribute information.

The proposed method, DPE, significantly outperforms all baseline models in handling unknown attributes. As shown in Table 1, DPE consistently achieves higher WGA across various datasets, with an average of 73.9%. This is a substantial improvement compared to ERM's average of 57.7%, and it also surpasses other robust methods such as DFR and RWY, which achieve 65.2% and 67.5%, respectively. On difficult datasets like CHEXPERT, DPE achieves a WGA of 76.8%, exceeding the performance of CRT (74.6%) and DFR (75.8%). Similarly, on the CELEBA dataset, which includes imbalances in hair color and gender, DPE attains a WGA of 84.6%, outperforming other baselines. Overall, the results demonstrate DPE's capability to implicitly discover subpopulations and capture different aspects of the data.

### 5.2. Subgroup Annotations Further Enhance DPE's Performance in Addressing Subpopulation Shift

When subgroup annotations are available in the validation set, models can leverage this information to improve their robustness, either by using group-balanced subsampling or

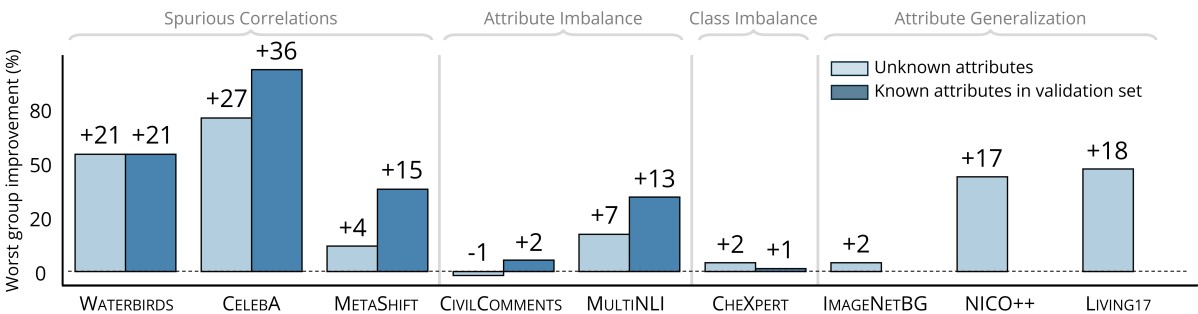

Figure 3. Worst-group improvement over ERM[*] when using DPE with and without subgroup annotations.

Table 2. Worst-group accuracy (WGA) on the test set with access to attribute annotations. The top half presents SubpopBench-style baselines and our method on a standard ERM backbone. The bottom half includes recent state-of-the-art methods and our method applied to a stronger ERM[*] backbone. DPE refers to our diversified prototypical ensemble. Group Info (Train/Val) indicates whether group labels are required: ✗ = no group info required, ✓ = group info required for hyperparameter tuning, ✓✓ = validation data required for both training and hyperparameter tuning. Best and second-best values within each half are **underlined and bolded** and **bold**, respectively. "-" indicates results not reported.

| Algorithm | Group Info (Train/Val) | WATERBIRDS | CELEBA | CIVILCOMMENTS | MULTINLI | METASHIFT | CHEXPERT |
|---|---|---|---|---|---|---|---|
| ERM | ✗/✗ | $69.1_{\pm4.7}$ | $57.6_{\pm0.8}$ | $63.2_{\pm1.2}$ | **$66.4_{\pm2.3}$** | $82.1_{\pm0.8}$ | $41.7_{\pm3.4}$ |
| CRT | ✗/✓ | $76.3_{\pm0.8}$ | $70.4_{\pm0.4}$ | **$68.5_{\pm0.0}$** | $65.4_{\pm0.1}$ | $83.1_{\pm0.0}$ | **$74.0_{\pm0.2}$** |
| ReWeightCRT | ✗/✓ | $76.3_{\pm0.2}$ | $71.1_{\pm0.5}$ | $68.2_{\pm0.4}$ | $65.3_{\pm0.1}$ | **$85.1_{\pm0.4}$** | $73.9_{\pm0.2}$ |
| DFR | ✗/✓✓ | **$89.0_{\pm0.2}$** | **$86.3_{\pm0.3}$** | $66.5_{\pm0.2}$ | $63.8_{\pm0.0}$ | $81.5_{\pm0.0}$ | **$75.4_{\pm0.6}$** |
| ERM + DPE | ✗/✓✓ | **$91.0_{\pm0.4}$** | **$87.7_{\pm0.6}$** | **$71.5_{\pm0.6}$** | **$74.8_{\pm0.3}$** | **$87.9_{\pm0.7}$** | — |
| ERM[*] | ✗/✗ | $77.9_{\pm3.0}$ | $66.5_{\pm2.6}$ | $69.4_{\pm1.2}$ | $66.5_{\pm0.7}$ | $80.0_{\pm0.0}$ | $75.6_{\pm0.4}$ |
| Group DRO | ✓/✓ | $91.4_{\pm1.1}$ | $88.9_{\pm2.3}$ | $70.0_{\pm2.0}$ | **$77.7_{\pm1.4}$** | — | — |
| RWG | ✓/✓ | $87.6_{\pm1.6}$ | $84.3_{\pm1.8}$ | **$72.0_{\pm1.9}$** | $69.6_{\pm1.0}$ | — | — |
| JTT | ✗/✓ | $86.7$ | $81.1$ | $69.3$ | $72.6$ | — | — |
| CnC | ✗/✓ | $88.5_{\pm0.3}$ | $88.8_{\pm0.9}$ | $68.9_{\pm2.1}$ | — | — | — |
| SSA | ✗/✓✓ | $89.0_{\pm0.6}$ | $89.8_{\pm1.3}$ | $69.9_{\pm2.0}$ | $76.6_{\pm0.7}$ | — | — |
| DFR* | ✗/✓✓ | $92.9_{\pm0.2}$ | $88.3_{\pm1.1}$ | $70.1_{\pm0.8}$ | $74.7_{\pm0.7}$ | — | — |
| GAP (Last Layer) | ✗/✓✓ | $93.2_{\pm0.2}$ | **$90.2_{\pm0.3}$** | — | $74.3_{\pm0.2}$ | — | — |
| GAP (All Layer) | ✗/✓✓ | **$93.8_{\pm0.1}$** | **$90.2_{\pm0.3}$** | — | **$77.8_{\pm0.6}$** | — | — |
| ERM[*] + DPE | ✗/✓✓ | **$94.1_{\pm0.4}$** | **$90.3_{\pm0.7}$** | $70.8_{\pm0.8}$ | $75.3_{\pm0.5}$ | **$91.7_{\pm1.3}$** | **$76.0_{\pm0.3}$** |

reweighting, particularly against underrepresented or challenging subpopulations. Table 2 presents a comparison of WGA for several methods across datasets where subgroup annotations are known. Compared to the scenario where attributes are unknown, all models show improved performance in this setting. For instance, CRT achieves a WGA of 70.4% on CELEBA and 68.5% on CIVILCOMMENTS, demonstrating a clear advantage over its performance when attributes are unknown. DFR and JTT demonstrate superior performance, achieving a WGA of 86.3% and 81.1% on CELEBA, respectively. DPE delivers the most substantial improvements across almost all datasets. As seen in Table 2, DPE achieves the highest average WGA at 83.0%, consistently outperforming other methods. On METASHIFT, DPE reaches a WGA of 91.7%, surpassing ReWeightCRT (85.1%) and DFR (81.5%). For WATERBIRDS, DPE attains a WGA of 94.1%, which is higher than GAP (93.8%) and DFR (89.0%). Furthermore, in CIVILCOMMENTS, a dataset

characterized by significant identity-based imbalances, DPE achieves a WGA of 70.8%, outperforming CRT and DFR.

### 5.3. DPE Mitigates Challenging Subpopulation Shift

One of the most critical challenges for machine learning models in subpopulation shift is the ability to handle attribute imbalance (AI) and attribute generalization (AG). As shown in the Yang et al. (2023), most existing methods—including well-known algorithms like GroupDRO and JTT—show limited improvement in scenarios involving AI and AG. Yang et al. (2023) highlights that none of the existing methods significantly outperform ERM in AG settings, indicating their inability to cope with the imbalanced/unseen attributes in the test data. In contrast, DPE consistently achieves the highest WGA in both AI (e.g. CHEXPERT, CIVILCOMMENTS) and AG (e.g. NICO++, LIVING17 (Fig. 3, Table 1). DPE's use of a prototypical ensemble

allows each prototype to explore substructures in the latent space without requiring knowledge about the number of subpopulations or the detection of instances in vulnerable subpopulations. The diverse prototypes ensure that the model does not rely on the most frequent attributes alone but instead learns a broader representation of the underlying data, resulting in better performance across all subgroups.

### 5.4. ERM vs. ERM*: Disentangling Backbone Effects

To isolate the impact of our prototypical ensemble from the underlying feature extractor, we compare four variants: ERM, ERM* (with stronger augmentation and longer training), and both combined with DPE. As shown in Tables 2a and 2b, DPE improves worst-group accuracy (WGA) consistently regardless of the backbone. For example, on WA-TERBIRDS, WGA increases from 69.1% (ERM) and 77.9% (ERM*) to 91.0% and 94.1% with DPE, respectively. Similar trends hold across datasets like CELEBA (57.6% → 81.9% with ERM+DPE; 66.5% → 84.6% with ERM*+DPE) and MULTINLI (66.4% → 69.3%; 66.5% → 70.9%). These consistent gains both with and without subgroup availability indicate that DPE's robustness stems from prototype diversification rather than representation quality alone, and that it serves as an effective, modular improvement atop standard training pipelines.

## 6. Ablation study

### 6.1. Prototype Diversification is Critical for Capturing Diverse Subpopulations

In this section, we investigate the impact of prototype diversity strategies on the performance of the prototypical ensemble, when subgroup annotation is available, specifically comparing three scenarios: (1) using a fixed subset of data for each prototype, (2) training with a random subset for each prototype, and (3) incorporating both random subset selection and inter-prototype similarity loss to maximize diversity across the ensemble members. Fig. 4 highlights the WGA across different datasets as we increase the number of prototypes under these scenarios.

In the first scenario, where a **fixed subset** is used for each prototype, the performance initially improves as we increase the number of prototypes. However, this improvement quickly plateaus, suggesting that the prototypes learn redundant features when trained on the same subset. Without introducing diversity, adding more prototypes beyond a certain point fails to significantly boost WGA.

The second scenario, involving **random subset selection**, introduces some level of diversity by ensuring each prototype is trained on a different random subset of data. This strategy leads to noticeable improvements (Fig. 4) as the number of prototypes increases, with each prototype focus-

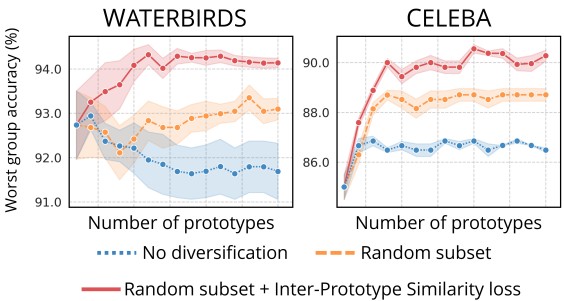

*Figure 4.* Effect of different ensemble diversification methods on performance with different numbers of ensemble members.

ing on different parts of the data distribution.

In the third scenario, we apply both **random subset selection** and **inter-prototype similarity loss (IPS)**, which explicitly minimizes the overlap between prototypes in latent space. This yields the best overall performance across all datasets, with substantial gains in WGA as the ensemble size increases. IPS ensures that each prototype independently contributes to the overall performance, reducing redundancy and enhancing the ensemble's ability to cover a broader range of data distributions with more prototypes.

### 6.2. Prototypical Ensembles Outperform Linear Ensembles

We compare the WGA of the prototypical ensemble with that of a linear ensemble, focusing on scenarios where both models have the same number of ensemble members and were trained on random subgroup-balanced subsets. Fig. 5 highlights the performance differences across multiple datasets, demonstrating the clear advantage of the prototypical ensemble over the linear ensemble when dealing with subpopulation shift. The **linear ensemble**, which consists of multiple independently trained linear classifiers, shows similar performance to the original DFR, which only trained three classifiers as the ensembles. On the other hand, from both Fig. 4 and Fig. 5, the **prototypical ensemble** shows steady improvement when increasing the ensemble size and consistently outperforms the linear ensemble across all datasets regardless of the attribute availability. Moreover, prototype ensemble markedly improves over linear ensemble on attribute imbalance (MULTINLI) and attribute generalization (IMAGENETBG, LIVING17), the two most challenging types of subpopulation shift.

### 6.3. Effect of Ensemble Size

We analyze the impact of the number of prototypes per class, denoted by $N$, on worst-group accuracy (WGA). To quantify the performance gain, we define the percentage

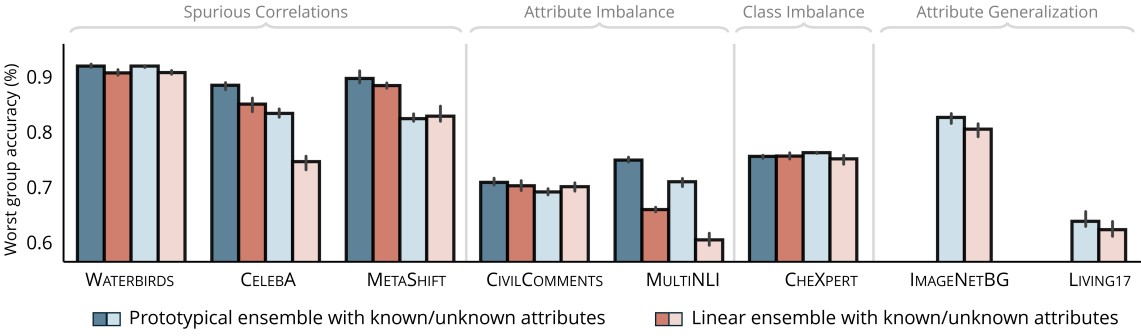

*Figure 5.* Linear ensemble versus prototypical ensemble with known and unknown attributes. NICO++ is not included in the plot since in our experiments, the worst-group accuracy of the linear ensemble on this dataset is zero.

improvement as $\Delta_N = \frac{\text{WGA}_N - \text{WGA}_1}{\text{WGA}_1} \times 100$, where $\text{WGA}_N$ denotes the worst-group accuracy when using $N$ prototypes per class. We conduct this ablation on four representative datasets: WATERBIRDS, CELEBA, METASHIFT, and CHEXPERT, evaluated under both known and unknown attribute settings.

Our results show that increasing $N$ leads to diminishing returns beyond a certain point. Specifically, we observe the following average improvements: $\Delta_5 = 2.4\%$, $\Delta_{10} = 3.3\%$, $\Delta_{15} = 3.7\%$, $\Delta_{25} = 3.7\%$, and $\Delta_{40} = 3.7\%$. These results indicate that WGA saturates around $N = 15$, and further increases in ensemble size result in minimal additional gain due to overlapping prototypes in latent space. These findings empirically support that our use of $N = 15$ balances robustness and computational efficiency.

### 6.4. Sensitivity to Hyperparameters

We assess the sensitivity of DPE to two core hyperparameters: the temperature $\tau$ and the inter-prototype similarity loss weight $\alpha$. Across $\tau \in \{10, 20, 30, 40\}$ and $\alpha \in \{10^4, 5 \times 10^4, 10^5, 5 \times 10^5\}$ on WATERBIRDS, METASHIFT, and LIVING17, performance varies within 1–2%, confirming DPE's robustness to hyperparameter tuning. The only exception is LIVING17, which shows greater variability due to its fine-grained subpopulation structure. See Appendix C.5 for the full analysis.

### 6.5. Runtime and Memory Overhead

To quantify computational efficiency, we benchmarked runtime and GPU memory usage on an RTX6000 with a ResNet-50 backbone and batch size 1. The results show that increasing the number of prototypes from 15 to 100 leads to a modest increase in per-batch latency (from 0.0031s to 0.0045s) and memory (from 0.20 GB to 0.85 GB). These results confirm that DPE is scalable and incurs minimal overhead even with large ensemble sizes.

### 6.6. Standard Accuracy and Robustness Trade-off

To evaluate whether robustness gains trade off with overall performance, we report average accuracy in Appendix C.6. DPE maintains competitive or improved accuracy compared to ERM and ERM\*, in both known and unknown attribute settings. For example, on METASHIFT, ERM\* + DPE improves WGA from 80.0% to 83.6% and average accuracy from 93.2% to 93.8%. These results support that DPE enhances fairness without sacrificing accuracy.

## 7. Limitations

A limitation of our approach is some additional complexity introduced by using an ensemble and some additional hyperparameters introduced by the method; however, this limitation is outweighed by improved performance and greater flexibility, as it does not require identification of sub-populations like JTT (Liu et al., 2021) and CnC (Zhang et al., 2022). The method remains computationally efficient as the ensemble is trained only on pre-extracted features, adding approximately two minutes per prototype on most datasets. Another limitation is the lack of a formal theoretical explanation for why prototype diversification improves worst-group accuracy. To partially address this, we conducted an exploratory analysis on WATERBIRDS dataset in which a third-party annotator (ChatGPT) examined samples closest to each prototype and found consistent semantic patterns such as habitat type and pose, suggesting that DPE may implicitly recover meaningful subgroups despite not using group labels (Appendix C.7). While promising, these findings are qualitative, and developing a theoretical understanding remains an open direction. Finally, similar to prior works, our method relies on ERM for feature extraction, which may underperform in low-data or weak-label settings; future work could explore integrating self-supervised learning to improve generalization in such regimes.

## Acknowledgements

This work was supported by the Canada CIFAR AI Chair program, the Vector Institute, and the Canada Research Chair Tier I in Medical Informatics. We gratefully acknowledge funding from the Canadian Institutes of Health Research (CIHR) and the Natural Sciences and Engineering Research Council of Canada (NSERC). Additional resources used in preparing this research were provided, in part, by the Province of Ontario, the Government of Canada through CIFAR, and the companies sponsoring the Vector Institute.

RGK is supported by a Canada CIFAR AI Chair and a Canada Research Chair Tier II in Computational Medicine.

## Impact Statement

This work introduces Diverse Prototypical Ensembles (DPE), a method designed to enhance model robustness under subpopulation shifts without relying on explicit group annotations. By leveraging class-balanced subsampling and inter-prototype diversification, DPE aims to mitigate performance disparities across subgroups, promoting fairness in model predictions. The potential societal impact of this work lies in its applicability to real-world scenarios where subgroup labels are unavailable or costly to obtain. DPE's ability to improve worst-group accuracy without significant trade-offs for majority groups suggests its utility in domains such as healthcare, finance, and education, where equitable model performance is critical. However, it is important to acknowledge that while DPE addresses certain fairness concerns, it does not eliminate all biases inherent in data or model architectures. Further research is necessary to assess DPE's effectiveness across diverse datasets and to explore its integration with other fairness-enhancing techniques.

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

# A. Subpopulation Shift Benchmark

## A.1. Datasets

WATERBIRDS (Wah et al., 2011) is a popular dataset for studying spurious correlations, where the background (water vs. land) is often confounded with the target label (waterbirds vs. landbirds). The key challenge in this dataset is that waterbirds are predominantly seen in water backgrounds, and landbirds are seen in land backgrounds. This creates a spurious correlation, making it difficult for models trained on such data to generalize to cases where birds appear in unusual backgrounds (e.g., landbirds in water environments). Models are evaluated on their ability to classify these atypical instances correctly.

*Table 1.* Summary of the WATERBIRDS dataset.

| Dataset | # Attr. | # Cls. | # Tr. | # Val. | # Test |
|---|---|---|---|---|---|
| WATERBIRDS | 2 | 2 | 4795 | 1199 | 5794 |

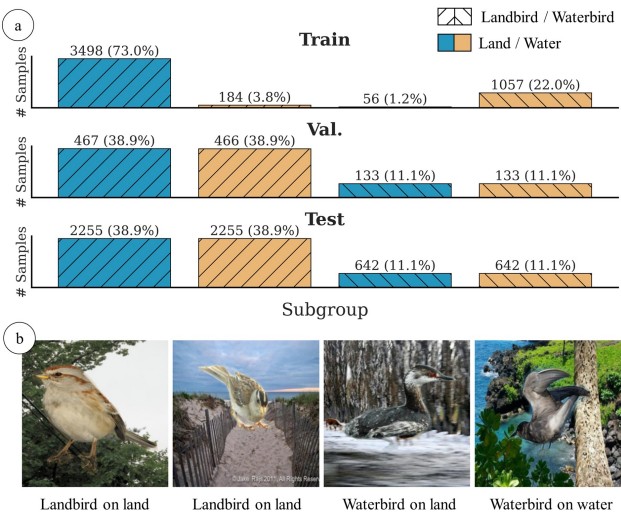

*Figure 1.* Class/Attribute distribution in WATERBIRDS dataset.

CELEBA (Liu et al., 2015) is a large-scale facial attribute dataset, widely used to explore the impact of attribute imbalance. The target task is to classify images based on hair color (blond vs. non-blond) with the confounding attribute being gender. Since blond hair is predominantly seen in females, models tend to overfit to this correlation, leading to poor performance on subgroups such as blond males or non-blond females. This dataset is used to test whether models can generalize across imbalanced attribute distributions and not rely solely on spurious associations.

*Table 2.* Summary of the CELEBA dataset.

| Dataset | # Attr. | # Cls. | # Tr. | # Val. | # Test |
|---|---|---|---|---|---|
| CELEBA | 2 | 2 | 162770 | 19867 | 19962 |

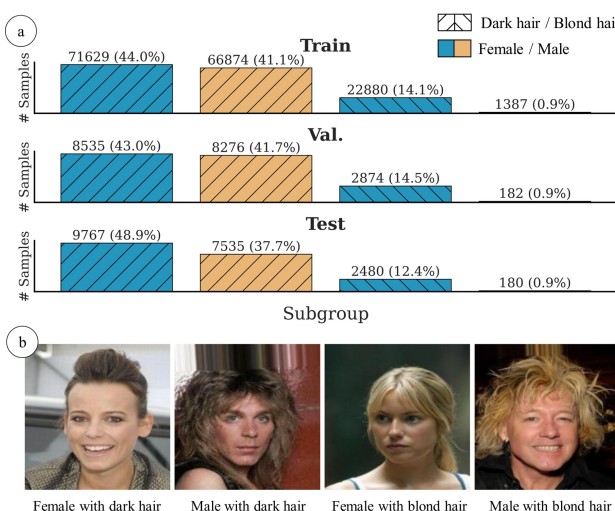

*Figure 2.* Class/Attribute distribution in CELEBA dataset.

CHEXPERT (Irvin et al., 2019) is a large-scale medical dataset consisting of chest radiographs. The dataset presents a significant class imbalance problem, with certain rare medical conditions being underrepresented in the training data. In this setting, models must be robust to imbalanced subpopulation distributions to perform well across both common and rare conditions. Evaluating models on this dataset demonstrates their ability to handle real-world medical tasks where some conditions may appear infrequently but are critical to correctly diagnose.

*Table 3.* Summary of the CHEXPERT dataset.

| Dataset | # Attr. | # Cls. | # Tr. | # Val. | # Test |
|---|---|---|---|---|---|
| CHEXPERT | 6 | 2 | 167093 | 22280 | 33419 |

CIVILCOMMENTS (Borkan et al., 2019) is a dataset collected from an online comment moderation platform, where the task is to predict toxicity in comments. This dataset is used to test robustness to identity-based subgroup shifts, as certain demographic groups (such as race or gender) are underrepresented in the data, and there is a strong risk of models developing biased predictions. The ability to correctly classify toxic comments across all identity groups is a key challenge in this dataset, and it provides a benchmark for testing fairness in machine learning models.

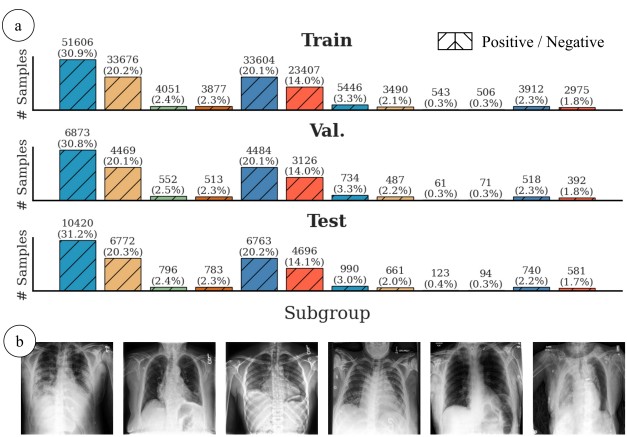

Figure 3. Class/Attribute distribution in CHEXPERT dataset. a) Each color represents one subgroup; b) Example images from the corresponding subgroups.

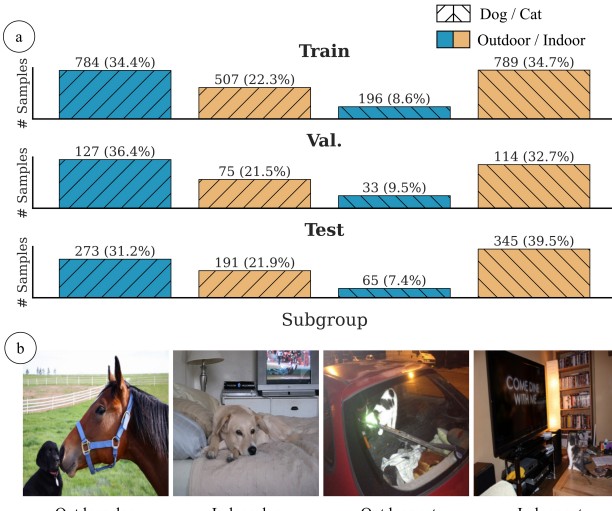

Figure 4. Class/Attribute distribution in METASHIFT dataset.

Table 4. Summary of the CIVILCOMMENTS dataset.

| Dataset | # Attr. | # Cls. | # Tr. | # Val. | # Test |
|---|---|---|---|---|---|
| CIVILCOMMENTS | 6 | 2 | 148304 | 24278 | 71854 |

MULTINLI (Schuhmann et al., 2022) is a natural language inference dataset where models must predict the relationship between pairs of sentences (entailment, contradiction, or neutral). The presence of linguistic artifacts, such as negation words (e.g., "no" or "never"), is often correlated with the contradiction label, creating a spurious association. This dataset is used to test whether models can avoid relying on such artifacts and instead generalize across more diverse sentence pairs.

Table 5. Summary of the MULTINLI dataset.

| Dataset | # Attr. | # Cls. | # Tr. | # Val. | # Test |
|---|---|---|---|---|---|
| MULTINLI | 6 | 2 | 206175 | 82462 | 123712 |

METASHIFT (Liang & Zou, 2022) is a dataset that introduces the concept of contextual distribution shifts. It contains various object classes (e.g., cats and dogs) in different contexts (e.g., indoor and outdoor environments). The task is to classify objects while being robust to changes in the context in which these objects are seen. This dataset is particularly useful for testing whether models can generalize to unseen combinations of objects and contexts during testing.

Table 6. Summary of the METASHIFT dataset.

| Dataset | # Attr. | # Cls. | # Tr. | # Val. | # Test |
|---|---|---|---|---|---|
| METASHIFT | 2 | 2 | 2276 | 349 | 874 |

IMAGENETBG (Xiao et al., 2021) is a modified version of ImageNet that emphasizes background noise as a source of variation in object recognition tasks. The challenge here lies in generalizing to new images where the background differs from what was seen during training. This dataset tests models' ability to disentangle foreground objects from background information, ensuring robustness to background shifts.

Table 7. Summary of the IMAGENETBG dataset.

| Dataset | # Attr. | # Cls. | # Tr. | # Val. | # Test |
|---|---|---|---|---|---|
| IMAGENETBG | N/A | 9 | 183006 | 7200 | 4050 |

NICO++ (Zhang et al., 2023) is another domain generalization dataset that contains various object categories with a focus on unseen attributes during testing. The key challenge in this dataset is that models must generalize to unseen combinations of object categories and attributes (e.g., "cat in autumn" or "dog on water"). It is an important benchmark for evaluating models' ability to handle attribute generalization and unseen variations in test data.

Table 8. Summary of the NICO++ dataset.

| Dataset | # Attr. | # Cls. | # Tr. | # Val. | # Test |
|---|---|---|---|---|---|
| NICO++ | 6 | 60 | 62657 | 8726 | 17483 |

LIVING17 (Santurkar et al., 2021) is part of the BREEDS benchmark, a dataset constructed to test hierarchical classification and domain generalization. In this dataset, the task is to classify living organisms into subcategories, and the challenge comes from unseen subclasses at the same

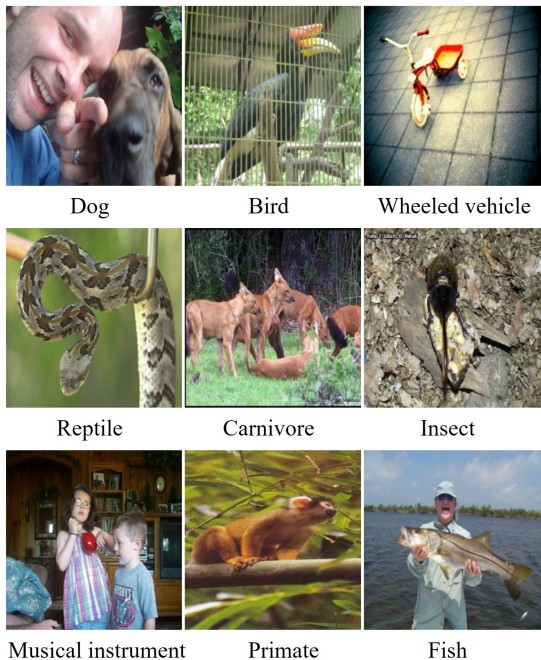

Dog       Bird       Wheeled vehicle

Reptile       Carnivore       Insect

Musical instrument       Primate       Fish

*Figure 5.* Example images from each class of IMAGENETBG dataset. Attributes are not available for this dataset.

hierarchical level during testing. This dataset assesses models' robustness in generalizing across hierarchical structures, where unseen subcategories must be correctly identified.

*Table 9.* Summary of the LIVING17 dataset.

| Dataset | # Attr. | # Cls. | # Tr. | # Val. | # Test |
|---------|---------|--------|-------|--------|--------|
| LIVING17 | N/A | 17 | 39780 | 4420 | 1700 |

## A.2. Evaluation metrics

### WORST-GROUP ACCURACY (WGA)

Worst-group accuracy measures the performance of a model on the subgroup of data where it performs the worst. Formally, let $\mathcal{G}$ denote the set of all groups, and let $\mathcal{D}_g$ represent the subset of data belonging to group $g \in \mathcal{G}$. Define the accuracy on group $g$ as:

$$\text{Acc}_g = \frac{1}{|\mathcal{D}_g|} \sum_{(x,y) \in \mathcal{D}_g} \mathbb{I}(f(x) = y),$$

where $f(x)$ is the model's prediction and $\mathbb{I}(\cdot)$ is the indicator function. The worst-group accuracy is then given by:

$$\text{WGA} = \min_{g \in \mathcal{G}} \text{Acc}_g.$$

This metric evaluates the robustness of the model to underrepresented or challenging groups in the dataset.

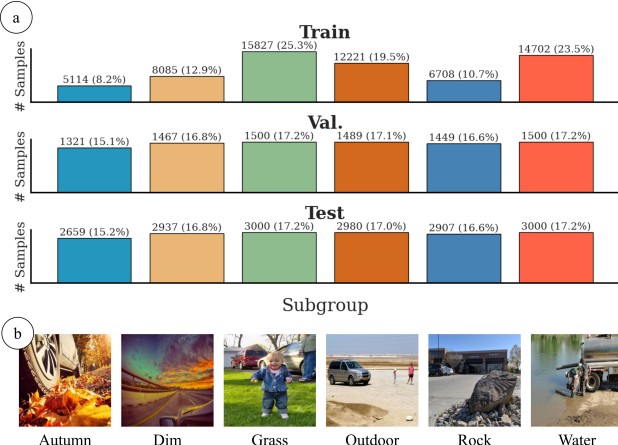

*Figure 6.* Attribute distribution in NICO++ dataset. a) Each color represents one subgroup; b) Example images from the corresponding subgroups.

### BALANCED ACCURACY (BA)

Balanced accuracy is designed to mitigate the effects of class imbalance by averaging the accuracy across all classes. Let $\mathcal{C}$ denote the set of all classes, and $\mathcal{D}_c$ the subset of data belonging to class $c \in \mathcal{C}$. The per-class accuracy is defined as:

$$\text{Acc}_c = \frac{1}{|\mathcal{D}_c|} \sum_{(x,y) \in \mathcal{D}_c} \mathbb{I}(f(x) = y).$$

The balanced accuracy is computed as:

$$\text{BA} = \frac{1}{|\mathcal{C}|} \sum_{c \in \mathcal{C}} \text{Acc}_c.$$

Balanced accuracy ensures that each class contributes equally to the overall evaluation, regardless of its frequency in the dataset.

### A.3. Baselines

**Empirical Risk Minimization (ERM)** serves as the standard training approach without any modifications for robustness. ERM optimizes for overall accuracy, often leading to suboptimal performance on underrepresented subpopulations, as it tends to prioritize the majority groups during training. While ERM is effective in balanced datasets, it has well-documented limitations in cases of spurious correlations or significant subpopulation shifts, making it a key point of comparison.

**Classifier Re-train (CRT, ReWeightCRT)** (Kang et al., 2020) focuses on first learning generalizable representations with standard sampling techniques. Then, in the second stage, the classifier is re-trained separately with class-balanced sampling or using non-parametric methods like Nearest Class Mean (NCM) or $\tau$-normalization to adjust

decision boundaries for better performance across underrepresented classes

**Deep Feature Reweighting (DFR)** (Kirichenko et al., 2022) aims to mitigate the impact of spurious correlations by decoupling feature learning from classifier training. DFR specifically retrains the classification layer on a balanced subset of the validation data, ensuring that learned features are leveraged in a more robust manner. These approaches offer improved performance on subpopulations but rely heavily on having access to validation data with known subgroups.

**Just Train Twice (JTT)** (Liu et al., 2021) proposes a two-stage approach to improve worst-group accuracy without requiring group annotations during training. In the first stage, JTT trains a standard ERM model and identifies the training examples it misclassifies, which often belong to groups affected by spurious correlations. In the second stage, JTT retrains the model by upweighting these misclassified examples to enhance the model's performance on the worst-performing groups. Despite not using group annotations during training, JTT significantly improves worst-group accuracy, achieving performance close to methods that do rely on group annotations, while maintaining competitive average accuracy across various datasets.

**Correct-n-Contrast (CnC)** (Zhang et al., 2022) designs a two-stage contrastive learning method to improve robustness against spurious correlations without requiring group labels during training. In the first stage, an ERM model is trained to infer spurious attribute labels by predicting groups that may correspond to spurious correlations. In the second stage, CnC uses contrastive learning to align the representations of same-class samples while ignoring spurious attributes. By sampling same-class instances with different spurious attribute predictions as positives, and different-class instances with the same spurious attribute predictions as negatives, CnC ensures that the learned representations focus on meaningful, class-specific features rather than spurious correlations.

**RWY** (Idrissi et al., 2022) presents a simple yet effective approach to improving worst-group accuracy through basic data balancing techniques such as subsampling and reweighting without requiring subgroup annotations. The method focuses on addressing group and class imbalances in datasets by either subsampling large groups/classes or reweighting samples to ensure balanced mini-batches during training. The results show that these simple data balancing techniques achieve competitive worst-group accuracy compared to more complex state-of-the-art methods. Additionally, the study emphasizes that while attribute information is crucial for model selection during validation, it is less important during training.

**Automatic Feature Reweighting (AFR)** (Qiu et al., 2023) enhances group robustness without requiring access to group labels during training. AFR operates in two stages: first, a standard ERM model is trained on the full dataset. Then, AFR retrains only the last layer of the model on a reweighting set, automatically giving higher weight to examples where the ERM model underperforms, which typically belong to minority groups.

**Group-Aware Priors (GAP)** (Rudner et al., 2024) proposes a novel method to improve robustness to subpopulation shifts by introducing data-driven priors that favor models capable of generalizing well across different subgroups. GAP requires only a small set of group-labeled data and uses this to construct a probabilistic prior distribution over model parameters. The method operates in two steps: first, a neural network is trained using ERM; second, the group-aware prior is used to finetune the model, either on the entire network or by retraining just the last layer. By incorporating this group-aware prior, the model places a higher probability on parameters that achieve strong performance on worst-case subgroups, significantly improving group robustness.

# B. Implementation details

The training procedure for DPE framework consists of two stages and is implemented with dataset-specific hyperparameters detailed in Tables 10 and 11. Hyperparameters were selected to optimize the WGA on the validation set, ensuring fair evaluation across different datasets.

## B.1. Representation Learning

In the first stage, a feature extractor is trained using ERM with dataset-specific configurations (Table 10). Image datasets such as WATERBIRDS, METASHIFT, and IMAGENETBG use the SGD optimizer with a learning rate of 1e-2 and a batch size of 128, while text-based datasets such as CIVILCOMMENTS and MULTINLI use BertAdam with a learning rate of 1e-4 and a batch size of 16. The number of training epochs varies by dataset, ranging from 4 for CIVILCOMMENTS to 300 for WATERBIRDS.

*Table 10.* Hyperparameters of the representation learning.

| Dataset | # Epochs | Optimizer | LR | Batch size |
|---|---|---|---|---|
| WATERBIRDS | 300 | SGD | 1e-2 | 128 |
| CELEBA | 50 | SGD | 1e-2 | 128 |
| CIVILCOMMENTS | 4 | BertAdam | 1e-4 | 16 |
| MULTINLI | 5 | BertAdam | 1e-4 | 16 |
| METASHIFT | 100 | SGD | 1e-2 | 128 |
| IMAGENETBG | 20 | SGD | 1e-2 | 128 |
| NICO++ | 100 | SGD | 1e-2 | 128 |
| CHEXPERT | 30 | SGD | 1e-2 | 128 |
| LIVING17 | 50 | SGD | 1e-2 | 128 |

## B.2. Prototypical Ensemble Learning

In the second stage, class/group-balanced subsets of the validation set are used to train the prototype ensemble. Each ensemble member is trained sequentially with inter-prototype similarity loss ($L_{\text{IPS}}$) to encourage diversity among prototypes. The coefficient $\lambda$ for $L_{\text{IPS}}$ varies by dataset and is listed in Table 11, with values such as $5 \times 10^5$ for WATERBIRDS and $1 \times 10^5$ for CIVILCOMMENTS. Most datasets, including WATERBIRDS, CELEBA, and CHEXPERT, utilize the SGD optimizer with a batch size of 256, while METASHIFT uses a smaller batch size of 64 due to its smaller dataset size.

*Table 11.* Hyperparameters of prototypical ensemble learning.

| Dataset | LR | Optimizer | Batch size | $\lambda$ |
|---|---|---|---|---|
| WATERBIRDS | 1e-3 | SGD | 256 | 5e5 |
| CELEBA | 5e-4 | SGD | 256 | 5e5 |
| CIVILCOMMENTS | 1e-4 | SGD | 256 | 1e5 |
| MULTINLI | 1e-4 | SGD | 256 | 1e5 |
| METASHIFT | 1e-2 | SGD | 64 | 1e5 |
| IMAGENETBG | 1e-3 | SGD | 256 | 5e5 |
| NICO++ | 1e-2 | SGD | 256 | 1e5 |
| CHEXPERT | 1e-3 | SGD | 256 | 5e5 |
| LIVING17 | 5e-5 | SGD | 256 | 1e5 |

# C. Additional Results

## C.1. Effect of Diversification Strategies on Ensemble Performance

Figure 7 extends the results shown in Figure 5 of the main paper to four datasets, illustrating the effect of different ensemble diversification strategies on both worst-group accuracy and balanced accuracy as the number of ensemble members increases. Three methods are compared: (1) fixed subset training (no diversification), (2) random subset training, and (3) random subset selection combined with inter-prototype similarity loss. The results show that the combined approach consistently achieves the highest worst-group accuracy across datasets. This underscores the importance of using both explicit and implicit diversification mechanisms to ensure that the ensemble captures a broad range of data distributions and subpopulation dynamics.

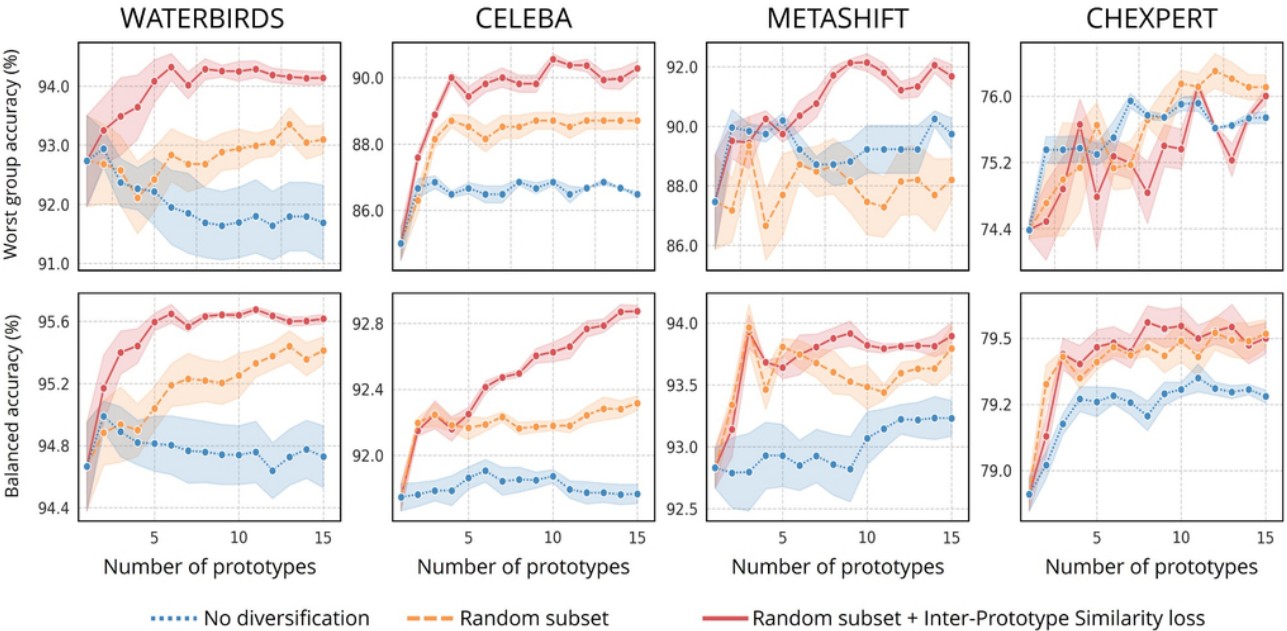

Figure 7. Effect of different ensemble diversification methods on performance with different numbers of ensemble members.

## C.2. Prototype Diversity Visualization

Figure 8 presents pairwise cosine similarity matrices for the first five prototypes within the ensemble, visualized separately for the "Landbird" and "Waterbird" classes. These matrices quantify the similarity between prototype embeddings, where values closer to 0 indicate greater diversity. The visualization demonstrates that our inter-prototype similarity loss effectively encourages representation diversity by reducing prototype overlap, enabling different prototypes to capture distinct subpopulation characteristics.

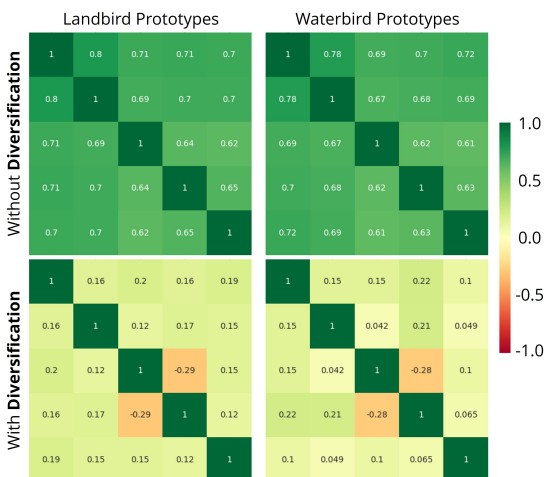

Figure 8. Pairwise cosine similarity matrices for the first 5 prototypes in the ensemble of the WATERBIRDS dataset.

## C.3. Runtime and Memory Efficiency

To assess the computational overhead introduced by our prototypical ensemble, we benchmarked DPE on an RTX6000 GPU using a ResNet-50 backbone and $k$=1000 classes, with a batch size of 1. Table 12 shows the runtime and memory usage as a function of the number of prototypes per class. We observe that inference time increases only slightly as the number of prototypes increases, from 0.0031s with 15 prototypes to 0.0045s with 100 prototypes. Memory usage grows more noticeably, increasing from 0.2032 GB to 0.8517 GB. In comparison, a standard linear classifier requires 0.0032s per batch and 0.104 GB of memory. These results suggest that DPE introduces minimal runtime overhead while remaining memory-efficient, even when using up to 100 prototypes, making it practical for large-scale applications.

*Table 12.* Time and memory benchmarking on RTX6000 with $k$=1000 classes (ResNet-50, batch size = 1).

| Model Head | # Prototypes | Time per Batch (s) | GPU Memory (GB) |
|---|---|---|---|
| DPE | 15 | 0.0031 | 0.2032 |
| DPE | 20 | 0.0033 | 0.2413 |
| DPE | 30 | 0.0033 | 0.3176 |
| DPE | 100 | 0.0045 | 0.8517 |
| Linear (Baseline) | N/A | 0.0032 | 0.1040 |

## C.4. Different Types of Ensemble-based Techniques

In this ablation study, we compare the performance of three ensemble learning strategies—Voting, Bagging, and Stacking—to the proposed ensemble approach, evaluating their efficacy in handling classification tasks under diverse conditions. The Voting ensemble aggregates predictions from heterogeneous base models, including Logistic Regression, Decision Tree, and Support Vector Machine (SVM), by employing both hard voting (majority rule) and soft voting (weighted probabilities). The Bagging ensemble utilizes Decision Trees as base learners and trains multiple instances on bootstrapped subsets of the training data, reducing variance and improving generalization. In contrast, the Stacking ensemble leverages a meta-learner (Gaussian Naïve Bayes) to integrate predictions from diverse classifiers (Logistic Regression, Decision Tree, and SVM), optimizing the final decision boundary through hierarchical learning. All models were trained on the same input features from the validation set used for training DPE. The group-balanced sampling strategy is used for WATERBIRDS and CELEBA, while class-balanced sampling strategy is used for LIVING17. Predictions were extracted from both the ensemble models and their individual base learners to analyze decision consistency and diversity. The results in this benchmark show that DPE consistently outperforms existing ensemble-based approaches under subpopulation shifts.

*Table 13.* Performance comparison of ensemble methods.

| Method | CELEBA | | WATERBIRDS | | LIVING17 | |
|---|---|---|---|---|---|---|
| | BAcc | WGA | BAcc | WGA | BAcc | WGA |
| Voting | 92.65 | 89.61 | 95.05 | 92.06 | 82.88 | 47.00 |
| Bagging | 90.87 | 81.11 | 94.01 | 92.06 | 75.71 | 34.00 |
| Stacking | 92.47 | 85.56 | 95.36 | 93.66 | 74.24 | 36.00 |
| DPE | **92.87** | **90.31** | **95.61** | **94.13** | **87.03** | **63.00** |

## C.5. Sensitivity Analysis

We evaluate the robustness of our method to two critical hyperparameters: the inverse temperature parameter $(1/\tau)$ used in the entropic similarity function, and the IPS loss coefficient $(\alpha)$ that controls the strength of prototype diversification. Table 14 presents the worst-group accuracy across multiple values of each hyperparameter, while Table 15 reports the corresponding overall accuracy. We observe that performance is generally stable across a wide range of settings for both hyperparameters. Notably, the method maintains high accuracy on WATERBIRDS and METASHIFT, with only moderate sensitivity observed on the more challenging LIVING17 dataset. These results suggest that the method does not require fine-tuning to achieve strong robustness and generalization across subpopulations.

*Table 14.* Sensitivity analysis of worst-group accuracy (WGA) to the inverse temperature ($1/\tau$) and the IPS loss coefficient ($\alpha$).

| Dataset | $1/\tau$ | | | | Summary | | $\alpha$ | | | | Summary | |
|---|---|---|---|---|---|---|---|---|---|---|---|---|
| | 10 | 20 | 30 | 40 | Mean | STD | 1e4 | 5e4 | 1e5 | 5e5 | Mean | STD |
| Waterbirds | 93.6 | 93.9 | 94.1 | 94.5 | 94.03 | 0.38 | 93.6 | 94.4 | 94.1 | 94.1 | 94.05 | 0.33 |
| MetaShift | 89.7 | 91.7 | 90.8 | 91.3 | 90.88 | 0.87 | 90.5 | 91.7 | 90.7 | 90.8 | 90.93 | 0.53 |
| Living17 | 63.0 | 58.7 | 57.3 | 55.3 | 58.58 | 3.26 | 63.0 | 61.7 | 61.7 | 62.0 | 62.10 | 0.62 |

*Table 15.* Sensitivity analysis of overall accuracy to the inverse temperature ($1/\tau$) and the IPS loss coefficient ($\alpha$).

| Dataset | $1/\tau$ | | | | Summary | | $\alpha$ | | | | Summary | |
|---|---|---|---|---|---|---|---|---|---|---|---|---|
| | 10 | 20 | 30 | 40 | Mean | STD | 1e4 | 5e4 | 1e5 | 5e5 | Mean | STD |
| Waterbirds | 96.4 | 96.2 | 96.0 | 95.9 | 96.13 | 0.22 | 96.3 | 95.9 | 96.0 | 95.9 | 96.03 | 0.19 |
| MetaShift | 93.8 | 93.7 | 93.9 | 93.9 | 93.83 | 0.10 | 93.9 | 93.9 | 93.7 | 93.7 | 93.80 | 0.12 |
| Living17 | 87.0 | 87.2 | 87.1 | 86.9 | 87.05 | 0.13 | 87.2 | 87.0 | 87.0 | 87.1 | 87.08 | 0.10 |

## C.6. Standard Accuracy

In addition to robustness metrics such as worst-group accuracy, we report the standard (average) accuracy of all evaluated methods across six benchmark datasets in Table 16. These results provide a complementary view of model performance, reflecting how well each method performs on the overall population rather than just on the most challenging subgroups. While some methods show strong gains in worst-group accuracy, they may trade off slightly in overall accuracy, particularly when subgroup reweighting is involved. Our proposed DPE method achieves competitive or superior average accuracy, especially when combined with a stronger ERM* backbone, indicating that robustness improvements do not come at the cost of general performance.

*Table 16.* Standard (average) accuracy for all methods across datasets. Group Info indicates whether group labels are used for training and/or validation: ✗/✗: No group info required, ✗/✓: Group info used for hyperparameter tuning, ✗/✓✓: Validation group labels required during training and tuning, ✓/✓: Full group labels required.

| Algorithm | Group Info | WATERBIRDS | CELEBA | CIVILCOMMENTS | MULTINLI | METASHIFT | CHEXPERT |
|---|---|---|---|---|---|---|---|
| ERM | ✗/✗ | 84.1±1.7 | 95.0±0.1 | 85.4±0.2 | 80.9±0.3 | 91.5±0.2 | 88.6±0.7 |
| CRT | ✗/✓ | 89.2±0.1 | 94.1±0.1 | 83.0±0.0 | 80.2±0.0 | 91.5±0.0 | 79.1±0.1 |
| ReWeightCRT | ✗/✓ | 89.4±0.3 | 94.2±0.1 | 83.4±0.0 | 80.2±0.0 | 91.3±0.1 | 79.0±0.0 |
| DFR | ✗/✓✓ | 92.2±0.2 | 91.2±0.1 | 81.3±0.0 | 80.2±0.0 | 90.5±0.4 | 78.9±0.2 |
| ERM + DPE | ✗/✓✓ | 92.5±0.2 | 89.8±0.2 | 82.2±0.2 | 81.3±0.2 | 91.2±0.1 | - |
| ERM* | ✗/✗ | 92.1±0.2 | 94.0±0.2 | 83.3±1.4 | 81.9±0.2 | 93.2±0.1 | 79.4±0.3 |
| Group DRO | ✓/✓ | 93.5 | 92.9 | 88.9 | 81.4 | - | - |
| RWG | ✓/✓ | - | - | - | - | - | - |
| JTT | ✗/✓ | 93.3 | 88.0 | 91.1 | 78.6 | - | - |
| CnC | ✗/✓ | 90.9±0.1 | 89.9±0.5 | 81.7±0.5 | - | - | - |
| SSA | ✗/✓✓ | 92.2±0.9 | 92.8±0.1 | 88.2±2.0 | 79.9±0.87 | - | - |
| DFR* | ✗/✓✓ | 94.2±0.4 | 91.3±0.3 | 87.2±0.3 | 82.1±0.2 | - | - |
| GAP (Last Layer) | ✗/✓✓ | 94.6±0.2 | 91.7±0.2 | - | 81.9±0.0 | - | - |
| GAP (All Layer) | ✗/✓✓ | 95.6±0.1 | 91.5±0.1 | - | 82.5±0.1 | - | - |
| ERM* + DPE | ✗/✓✓ | 96.0±0.1 | 91.9±0.3 | 81.6±0.2 | 81.6±0.2 | 93.8±0.5 | 79.0±0.2 |

## C.7. Exploratory Analysis of Prototype-Subgroup Alignment

To gain insight into the semantic structure captured by the Diversified Prototypical Ensemble (DPE), we performed an exploratory analysis on the WATERBIRDS dataset. For each prototype, we retrieved the top-10 closest validation samples and used ChatGPT to interpret the emerging patterns. As shown in Figures 9 and 10, the learned prototypes exhibit consistent alignment with ecologically or visually meaningful subpopulations, even though no subgroup labels were used

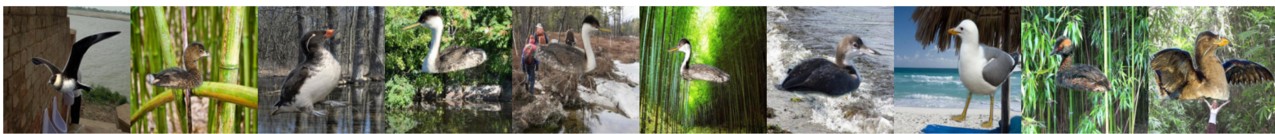

**"Diving and surface-swimming waterbirds in open aquatic environments"**
This prototype captures ecologically coherent waterbirds — mostly **grebes, loons, cormorants, and gulls** — all adapted to **diving or floating** on water. The visual consistency in **plumage contrast, aquatic setting, and pose** makes this a prototypical representation of the waterbird class.

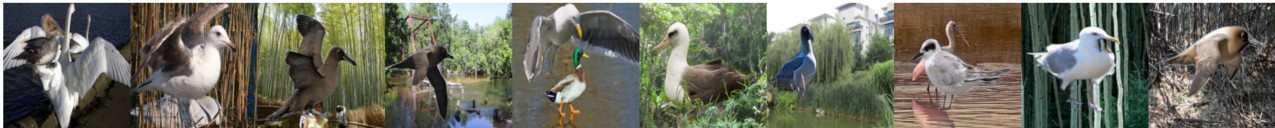

**"Large-bodied waterbirds in extended flight or lift-off poses"**
Prototype 2 groups waterbirds with **broad wingspans**, often shown **in flight or preparing for takeoff**, including albatrosses, gulls, and terns. While backgrounds vary (bamboo, docks, ponds), the shared latent theme centers on **aerodynamic posture and silhouette geometry**, forming a distinct subpopulation that reflects a **kinetic visual mode** of aquatic birds.

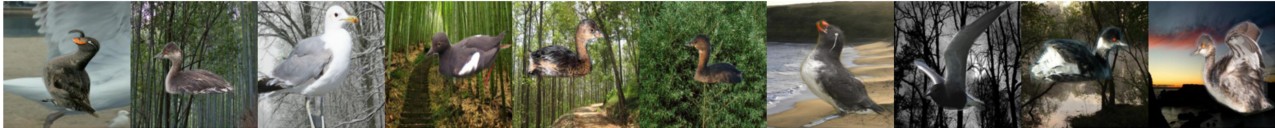

**"Compact-bodied diving birds with rounded silhouettes in terrestrial or ambient-light settings"**
This prototype groups **grebes, auklets, guillemots**, and terns — birds that share a **morphologically compact, rounded form** with low profiles and dark plumage. Despite their aquatic nature, they're often shown in **mismatched land-based or bamboo-heavy scenes**, suggesting a **visually-coherent but ecologically-confounded** latent cluster.

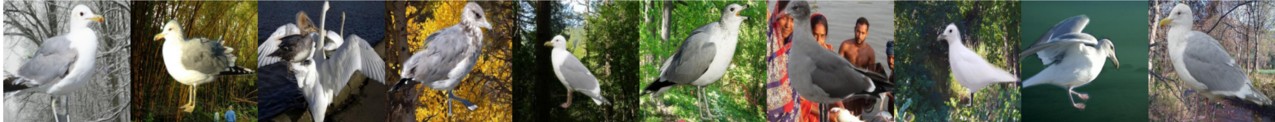

**"Standing gulls in upright posture with clean visual separation"**
This prototype forms a highly coherent group of **California, Glaucous-winged, Heermann's, and Ring-billed Gulls**, unified by their **gray-and-white plumage**, **yellow bills**, and **erect standing poses**. It tolerates a wide range of **backgrounds**, but emphasizes **pose uniformity and size**. A single **Hooded Merganser outlier** appears, likely due to **visual mimicry in shape** rather than ecological or taxonomic alignment.

*Figure 9.* Waterbird Prototypes. Each row depicts the top-10 validation samples closest to one of the prototypes learned for the WATERBIRDS class. Using ChatGPT for cluster interpretation, we observe that the prototypes induce structured prototype-subgroup alignment that meaningfully reflects bird morphology, pose, and context: divers and floaters in aquatic settings, large-bodied birds in dynamic flight poses, compact-bodied seabirds shown in terrestrial or bamboo-heavy scenes (an ecologically confounded but visually coherent mode), and upright-standing gulls with consistent visual separation.

during training. In the WATERBIRDS class, prototypes capture distinctions such as aquatic divers, large-bodied birds in dynamic flight, and seabirds in bamboo-heavy or terrestrial contexts. In the LANDBIRDS class, clusters emerge that reflect postural cues, background settings, and potential spurious correlations, such as songbirds appearing in human-associated environments. These findings suggest that DPE implicitly encourages subgroup discovery through diversification, which may contribute to its strong worst-group performance.

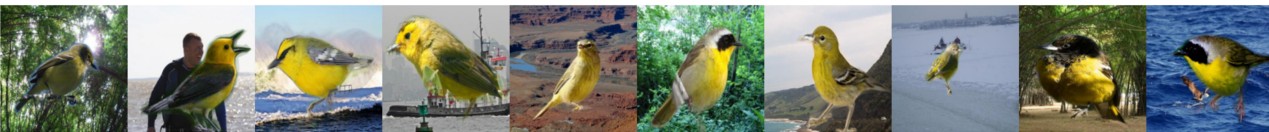

**"Small yellow warblers and vireos in varied habitats"**
This prototype captures a **highly color-consistent** group of small yellow songbirds, largely comprising **warblers, vireos, and flycatchers**. While visually coherent in terms of plumage and posture, the birds are shown in **diverse or mismatched backgrounds**, suggesting the prototype focuses on **foreground semantic structure** rather than context. It represents a meaningful **intra-class mode** for landbirds, especially the yellow-dominated cluster.

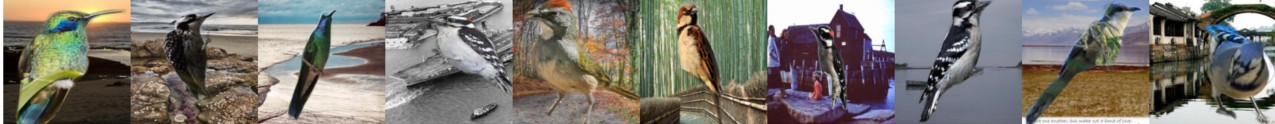

**"Upright-postured forest and edge-dwelling birds with structured or artificial backdrops"**
This prototype captures vertically oriented birds (especially woodpeckers and similar species) in settings that lack dense greenery — often featuring man-made structures, open water, docks, or stylized vertical environments (e.g. bamboo). It may encode a shape-surface or pose-context association, blending natural and built environments.

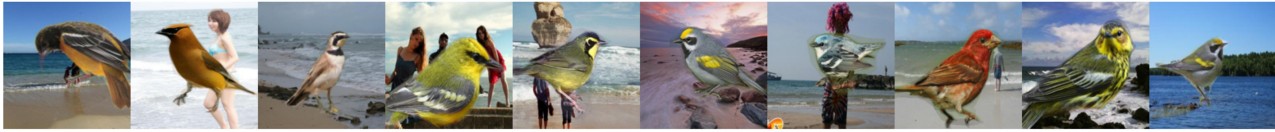

**"Forest songbirds in atypical coastal or human-present scenes"**
Prototype 4 clusters birds whose ecological identity is forest/shrubland but whose **visual context is dominated by beach, water, or people**. It captures a subpopulation prone to **spurious background correlations**, making it a key prototype for understanding failure under subpopulation shift.

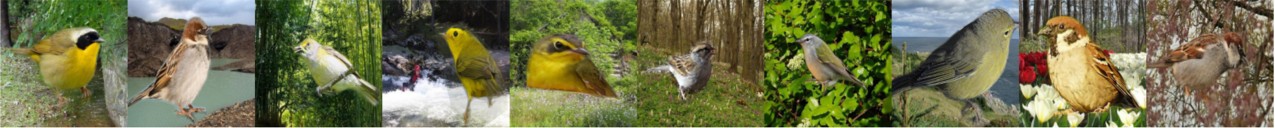

**"Muted-tone songbirds in dense green vegetation"**
This prototype captures small passerines — mainly warblers, vireos, and sparrows — that share a **camouflaged color scheme** and are seen in **leafy, ground-rich vegetation**. It reflects a low-visibility subpopulation that trades high contrast for ecological realism, possibly helping the model generalize across scenes with **low texture saliency**.

*Figure 10.* Landbird Prototypes. Each row shows the top-10 validation samples closest to one of the prototypes from our Diversified Prototypical Ensemble (DPE) model on the LANDBIRDS class of the WATERBIRDS dataset. ChatGPT was used to analyze each prototype's semantic structure. The results reveal that learned prototypes align with coherent visual or ecological subpopulations, for example, small yellow songbirds across diverse backgrounds, upright-postured forest-edge dwellers, forest birds appearing in beach or human-present scenes (highlighting spurious correlations), and muted-tone songbirds in leafy, texture-poor vegetation.

# D. Algorithm

Our training pipeline (**Algorithm** 1) consists of two stages: feature extractor training and prototypical ensemble training. In Stage 1, the feature extractor and classification head are trained using ERM on the training data to optimize feature representations. In Stage 2, an ensemble of class-specific prototypes is initialized and trained on class-balanced subsets or group-balanced subsets of the validation data, depending on the availability of the subgroup annotations. A distance-based loss and an inter-prototype similarity loss are used to update each ensemble member. During inference, class probabilities are computed using the joint decision of the members in the prototypical ensemble.

---

**Algorithm 1** Subpopulation Prototypical Ensemble Diversification

---

**Input** : Training data $D_{\text{train}}$, Validation data $D_{\text{val}}$, Number of ensemble members $N$, Diversity weight $\alpha$, Temperature $\tau$
**Output** : Trained feature extractor $f$, Ensemble of prototypes $\{p_j^{(i)}\}_{j=1}^N$, $i = 1, \ldots, K$

**Stage 1: Train Feature Extractor**
Initialize feature extractor $f$ and classification head $g$ **for** *each minibatch* $(X_{batch}, y_{batch}) \in D_{train}$ **do**
  Compute logits $z = g(f(X_{\text{batch}}))$ Compute loss $\mathcal{L}_{\text{CE}} = - \sum_i y_{\text{batch}}^{(i)} \log \sigma(z^{(i)})$ Update $f$ and $g$ to minimize $\mathcal{L}_{\text{CE}}$

**Stage 2: Train Prototypical Ensemble**
Initialize ensemble of prototypes $\mathcal{P} = \{\}$
**for** $j = 1$ **to** $N$ **do**
  Sample class-balanced subset $D_{\text{sub}}^{(j)}$ from $D_{\text{val}}$ Initialize prototypes $\{p_j^{(i)}\}_{i=1}^K$ with $p_j^{(i)} \sim \mathcal{N}(0, 0.01^2)$
  **for** *each minibatch* $(X, y) \in D_{sub}^{(j)}$ **do**
    Compute features $x = f(X)$ Compute distances $D(x, p_j^{(i)})$ using Equation (5) Compute loss $\mathcal{L}$ using Equation (6)
    **if** $j > 1$ **then**
      **Update** ensemble $\mathcal{P}$ with prototypes $\{p_j^{(i)}\}_{i=1}^K$ Compute inter-prototype similarity loss $\mathcal{L}_{\text{IPS}}$ (Equation (8)) using $\mathcal{P}$ and
      $\{p_j^{(i)}\}$ Set $\mathcal{L}_{\text{total}} = \mathcal{L} + \alpha \cdot \mathcal{L}_{\text{IPS}}$
    **else**
      Set $\mathcal{L}_{\text{total}} = \mathcal{L}$
    Update prototypes $\{p_j^{(i)}\}_{i=1}^K$ to minimize $\mathcal{L}_{\text{total}}$
  **Update** ensemble $\mathcal{P}$ with the newly trained prototypes $\{p_j^{(i)}\}_{i=1}^K$

**Inference**
Given test input $X$ Compute feature $x = f(X)$ **for** $i = 1$ **to** $K$ **do**
  Compute ensemble probability:
$$P(y = i | X) = \frac{1}{N} \sum_{j=1}^N \frac{\exp\left(- D(x, p_j^{(i)})\right)}{\sum_{k=1}^K \exp\left(- D(x, p_j^{(k)})\right)}$$
Predict label $\hat{y} = \arg\max_i P(y = i | X)$

---

