# OpenReview forum: "Diverse Prototypical Ensembles Improve Robustness to Subpopulation Shift"
_ICML.cc/2025/Conference — ICML 2025 poster_

### Official Review · Reviewer_mdbr · 2025-03-11

**Overall Recommendation:** 4

**Summary:**

The paper introduces **Diversified Prototypical Ensemble (DPE)** to enhance robustness against subpopulation shifts. The method trains multiple diverse prototypes per class on top of a frozen feature extractor and enforces feature diversity through **inter-prototype similarity (IPS) loss**. By restructuring the feature space, DPE ensures that minority subgroups receive dedicated representations, improving worst-group accuracy (WGA). Experiments across datasets like **Waterbirds and CelebA** show that DPE **outperforms prior reweighting-based approaches** in handling subpopulation imbalances. **Ablation studies confirm that prototype diversification and ensemble strategies drive these gains.** While effective, the authors highlight limitations in **prototype selection strategies and computational efficiency**, which require further optimization across diverse datasets.

**Claims And Evidence:**

1. DPE improves WGA compared to re-weighted approaches. Tables 1 and 2 support this claim.
 2. Figure 5 demonstrates that IPS loss encourages diverse feature representations.

**Essential References Not Discussed:**

Since prior work (e.g., hierarchical learning methods) has addressed subpopulation shift by structuring feature space, this paper should acknowledge that conceptual and ideally experimental overlap. While hierarchical classification relies on predefined label hierarchies, prototypical classifiers structure feature space through learned prototypes. Although implementation details differ, both approaches share the goal of improving generalization under subpopulation shifts, making the connection relevant. This point becomes as a novelty issue for this approach.

1. "Encoding Hierarchical Information in Neural Networks Helps in Subpopulation Shift", IEEE Transactions on Artificial Intelligence, 2023/ Fine Grained Visual Categorization (FGVC9) workshop, CVPR 2022.

**Experimental Designs Or Analyses:**

The experimental section overall confirms DPE improves WGA against previous approaches.

1. For gap in analysis, please refer to the experiments related to Non-living 26 or Entity 30 and feature visualizations via t-SNE.
2. A major issue is the effect of $ERM^(*)$ as feature extractor and not ERM. Experimental validation is required here.
3. Please provide standard accuracy as well. Without that, it is difficult to understand the drop for WGA.

These experiments are missing and will help in understanding the process.

**Methods And Evaluation Criteria:**

1. While DPE improves WGA, interpret ability analysis is lacking. If t-SNE or feature visualization plots be shared then it will add to the understanding.

2. DPE builds on $ERM^*$ and not on ERM. The advanced data augmentations affects learned features in an ERM setting as shown in works such as CutMix. DPE’s effectiveness on standard ERM should be tested to separate augmentation effects from prototype diversification.

3. How are N prototypes determined? An ablation on N across datasets would clarify its optimal selection.

4. Only Living-17 is used as one of the BREEDs benchmarks. Other datasets such as Non-living 26 and Entity-30 should be addressed as well. These are subpopulation shift vision benchmarks. One can use datasets such as CELEBA as a benchmark, but evaluation on either on Non-lIving 26 or Entity 30 is required.

**Other Comments Or Suggestions:**

1. Why is ERM and ERM* on LIVING-17 different from the original BREEDs paper?
2. Please explain the numbers in Figure 3. In Table 2, gap between ERM* and DPE is aorund 10% for LIVING-17. Why is it showing 18+ on Figure 3?

3. For Figure 5. what is the number of prototypes?

**Other Strengths And Weaknesses:**

1. Figure 2 is very well illustrated and conveys the goal of the paper.
2. Limitations are clearly explained.

**Questions For Authors:**

Please address the gaps in experiments provided under Experimental Designs and Analyses.
1. Results on Non-living 26
2. Results of ERM plus PDE on any 2 datasets

This should help us understand the efficacy of approach.

**Relation To Broader Scientific Literature:**

Very relevant for domain shift literature. Assuming subpopulation shift falls as a sub problem under domain shift.

**Theoretical Claims:**

No theoretical claims made.

---

> ### Author Rebuttal · Authors · 2025-04-01
>
> Thank you for your detailed and constructive feedback on our submission. We appreciate the time you have dedicated to evaluating our work, and we are pleased that you recognize the strength of our method in improving worst-group accuracy (WGA) under subpopulation shifts. Your concerns are
> - (1) Unclear contribution of DPE versus augmentations in ERM*
> - (2) Need for ablations on the number of prototypes
> - (3) Missing evaluation on other BREEDS datasets
> - (4) Absence of standard accuracy reporting
> - (5) A missing reference to prior work involving hierarchical learning
> - (6) Lack of interpretability analysis (e.g., t-SNE).
>
> We address these points below with new experiments, clarifications, and discussion. Full tables are provided at https://github.com/anonymous102030411/anon.
>
> ### 1. ERM vs. ERM∗ (Backbone Confounding Issue):
> To directly evaluate whether DPE’s performance gains stem from its architecture or stronger backbones, we have retrained the feature extractors for all datasets using the original ERM configurations from Yang et al. (2023)—without additional augmentations or extended training. The results using DPE on these standard ERM features are now included as ERM + DPE in the revised Table 2 and Table 3, (see [Tables 2-3](https://shorturl.at/Dy4Ab)). Across all datasets, DPE still achieves substantial gains over baselines under the same backbone. For example, on Waterbirds, ERM yields 69.1±4.7 WGA, while ERM + DPE achieves 91.0±0.5. Similarly, on CivilComments, DPE improves WGA from 63.2±1.2 (ERM) to 71.5±0.6. **These findings confirm that the improvement comes from DPE itself, not just stronger pretraining**. The updated results are detailed in our response to reviewer [VrPG](https://openreview.net/forum?id=qUTiOeM57J&noteId=JePMn0uimT).
>
> ### 2. Prototype Count ($N$) Ablation
> Following the reviewer’s suggestion, we have expanded our ablation on the number of prototypes N, which will be included in the appendix. For a more detailed discussion of our analysis, we direct the reader to the response to reviewer ___rN4y___.
>
> ### 3. Additional BREEDS Evaluation – Non-living 26 and Entity-30
> We thank the reviewer for this suggestion. BREEDS benchmarks such as Living-17, Non-living-26, and Entity-30 are important for studying subpopulation shifts as they construct domain splits based on fine-grained taxonomies, inducing subgroup variation. To address the comment, we conducted new experiments on Non-living-26 and Entity-30 using ERM and ERM + DPE, with and without ImageNet pretraining. Results show that DPE consistently improves worst-group accuracy (e.g., +3%–10%) across both source and target domains (see [Tables 4-5](https://shorturl.at/yuYnY)), These results validate DPE's robustness and generalization under complex subpopulation shift settings and will be included in the final version of the paper.
>
> ### 4. Reporting Standard Accuracy
> We thank the reviewer for highlighting the importance of reporting standard (average) accuracy to better contextualize the gains in worst-group accuracy (WGA). In response, we have included average accuracy results across all benchmarks in the appendix of the revised version. The average accuracy table corresponding to Table 2 is shown in [Table 6](https://shorturl.at/4CFih). The results demonstrate that both ERM + DPE and ERM* + DPE maintains comparable or improved average accuracy relative to ERM and ERM* across all datasets. These results confirm that the improvements in worst-group accuracy offered by DPE do not come at the cost of overall accuracy. On the contrary, DPE often improves both metrics. We include these results and observations in the revised manuscript.
>
> ### 5. Hierarchical Learning Literature
> We thank the reviewer for highlighting the connection to hierarchical representation methods. While DPE does not rely on explicit taxonomies, we agree it shares the goal of feature space structuring. We now cite and discuss "Encoding Hierarchical Information in Neural Networks Helps in Subpopulation Shift" in the Related Work section. We clarify that unlike those methods, DPE infers structure from data without pre-defined hierarchies, enabling application to a broader set of tasks lacking such annotations.
>
> ### 6. Feature Interpretability via t-SNE and Prototype Visualization
> To clarify the intuition behind the prototype ensemble’s ability to capture semantically meaningful and generalizable subpopulation features, we note that Figures 1.4 and 1.5 are real T-SNE visualizations from our Waterbirds experiments. They highlight prototypes aligned with semantic concepts (e.g., “small yellow bird”) rather than spurious ones (e.g., “land background”). We will include full-size T-SNE plots and additional examples from other datasets in the appendix.
>
> ---
>
> **Thank you for taking the time to review our work. If our answers resolve your concerns, we’d appreciate your consideration in raising the score. We're happy to clarify any remaining questions.**

---

> > ### Comment · Reviewer_mdbr · 2025-04-04
> >
> > I confirm that I have read the author rebuttal to my review. I acknowledge the response to my comments. The experiments are detailed.
> >
> > 1. I still do not see performance improvements for BREEDs benchmarks. For any subpopulation shift experiment, ImageNet pre-training based experiments are technically incorrect. The networks are already trained on features on which the shift happens. So I believe any ImageNet pre-trained experimental results should not be considered as valid results.
> >
> > 2. The positive result is that DPE does have benefits along with ERM as well and not only ERM*.
> >
> > I am increasing my score.

---

> > > ### Author Response · Authors · 2025-04-08
> > >
> > > We appreciate your follow-up and the score increase. Your feedback helped address the gap in our evaluation.
> > >
> > > We agree that ImageNet pre-trained models should not be used for the BREEDS benchmarks, and we will exclude the corresponding results in the revision. That said, our method consistently improves worst-group accuracy over the ERM baseline across all BREEDS datasets without ImageNet pre-training. We'll discuss the challenge of this benchmark in the final version.

---

### Official Review · Reviewer_aEu5 · 2025-03-14

**Overall Recommendation:** 4

**Summary:**

This paper studies the subpopulation shifting problem. To alleviate the issue, motivated by the idea of ensemble learning, the author proposes using a mixture of diversified prototypical classifiers over the feature prototypes of the subpopulations to classify different subpopulations correctly. Extensive experiments have been conducted on standard datasets. The author provides a comprehensive comparison of the proposed methods over the state-of-the-art method and justifies the effectiveness of the proposed method.

**Claims And Evidence:**

The paper's central claim is that explicitly encouraging diversity of the prototype-based predictors for each class could encourage subsequent ensemble members to capture the different decision boundaries for each subgroup, leading to better performance on classification under subpopulation shifting. The claim is reasonable, and the author provides extensive empirical studies to justify its effectiveness. Figure 2 provides a concise and clear demonstration of the motivation of the claim and the proposed method. The ablation study and error analysis further demonstrate that the proposed methods improve the classification performance as expected.

**Essential References Not Discussed:**

Paper [1] has not been cited/discussed in the paper. [1] tackles the subpopulation shifting problem in an incremental learning manner. In [1], the author proposes a prototype-based incremental learning method based on the generalized boosting theory [2,3] to learn new classifiers for novel subpopulations incrementally and combine the old and new classifiers sequentially. The insight of [1] is to calibrate the decision boundary over old and new subpopulations to correctly classify different subpopulations. Such a fine-grain modification on decision boundary is achieved by optimizing the margin-enforce loss [2,3], which, in theory, is equivalent to performing the ensemble learning via gradient-boosting that minimizes the residual error of the previously learned classifier.

The reviewer recognizes that the present submission is not a continual or incremental learning paper. However, considering that both [1] and the present submission have the same research goal of addressing subpopulation shifting and similar insight on alleviating the subpopulation shift issue by ensemble learning and prototype-based methods, the reviewer believes the author must provide sufficient discussion of the current submission and [1] to properly acknowledge the existing literature and inspire the boarder research community.

[1] Balancing between forgetting and acquisition in incremental subpopulation learning. ECCV 2022

[2] Multiclass boosting: margins, codewords, losses, and algorithms. JMLR 2019

[3] Boosting: Foundations and Algorithms. Kybernetes 2013

**Experimental Designs Or Analyses:**

The experimental designs and analysis are sound.

**Methods And Evaluation Criteria:**

Strengths:

Overall, the paper is well-written and well-motivated. The proposed prototype-based method is relevant and sound, and the idea of ensemble diversified prototype-based predictors to alleviate the subpopulation shifting issue is originated from the ensemble learning perspective but with further consideration of the subpopulation shifting problem. Extensive experiments have been conducted to verify the effectiveness of the proposed methods from different perspectives.


Weaknesses:

1. In line 220, the author chooses `N` prototypes per class for `K` classes. However, the author does not elaborate on how the `N` is being chosen, and the impact of the choice of `N` for training performance is unclear. The author may want to provide further ablation studies to demonstrate it, as the choice of `N` should be highly relevant to whether we can exactly achieve diversified predictors in practice, e.g., when `N` is not large enough, it is hard to achieve diversified predictors as there is not sufficient prototype to represent the feature space of the subgroups.

2. The cost of the proposed method has not been discussed. As the proposed method is prototype-based, the author must save the prototype and predictors to perform inference in test time. Given that the compared methods are not prototype-based, the author needs to provide the computational and storage costs between the proposed methods and the compared methods to let the reader know the trade-off between performance and the storage and computation cost among different kinds of methods to demonstrate the practicality of the proposed method.

3. The missing relevant literature in subpopulation shifting. Please refer to the Essential References Not Discussed for details.

**Other Comments Or Suggestions:**

N/A.

**Other Strengths And Weaknesses:**

N/A.

**Questions For Authors:**

N/A.

**Relation To Broader Scientific Literature:**

The submission is specific to addressing the subpopulation shifting problem and may not significantly impact broader scientific literature.

**Theoretical Claims:**

This is not a theory paper, and there is no theoretical analysis.

---

> ### Author Rebuttal · Authors · 2025-04-01
>
> Thank you for your positive and constructive review. We appreciate your recognition of our well-motivated method, the novel application of prototype-based ensemble for subpopulation shift, and the thorough experimental validation.
> Your main concerns include
> - (1) lack of ablation on the number of prototypes ($N$)
> - (2) missing analysis of computational and storage cost
> - (3) absence of discussion on related work in incremental subpopulation learning and boosting-based prototype methods.
>
> We address each of these points below with new results, expanded discussion, and clarifications. Full tables are provided at https://github.com/anonymous102030411/anon.
>
> ### 1. Choosing the Number of Prototypes
> We thank the reviewer for pointing out the importance of prototype count in achieving effective diversification. This is indeed a key design parameter in our method, and we have now conducted an extended ablation study to better quantify its effect on model performance. For a discussion of this study, we direct the reader to our response to reviewer ___[rN4y](https://shorturl.at/LBOK6)___.
>
> ### 2. Computation Cost and Complexity
> We agree that evaluating memory and computational requirements is essential. The additional cost of DPE relative to baselines is minimal, as our method only introduces $N$ prototypes per class ($D$-dimensional). In our experiments, with $K=10$ to $K=100$, $D=1024,$ and $N=10$ to $20$, this overhead is negligible compared to the backbone encoder.
> Inference speed is largely unaffected, as matching features against prototypes adds minimal cost relative to the encoder’s forward pass. While prototype storage scales with $K$, our experiments confirm that memory and compute demands remain manageable in practical settings. We refer the reviewer to our response to ___[rN4y](https://shorturl.at/LBOK6)___, where we provide detailed benchmarking results, including inference speed and GPU memory usage (see **[Table 1](https://shorturl.at/KekLM)**). We will update the manuscript to clarify that computation cost remains low and include the full benchmarking table in the appendix.
>
> ### 3. Addition of Relevant Citations
> We appreciate the reviewer’s suggestion and acknowledge that [1] addresses subpopulation shifts in an incremental learning setting. It employs an ERM-trained feature extractor and incrementally updates the classifier by combining previous and newly trained classifiers, with margin-enforce loss focusing on hard examples. The update balance is determined by measuring prototype distortion under the new classifier.
> Our work differs from [1] as follows:
> - Subpopulation characteristics: While [1] assumes distinct, predefined subpopulations, our method operates without prior knowledge of subpopulation structures and successfully generalizes to unknown shifts (as shown in Table 1, main paper).
>  - Training setup: We focus on distribution shift robustness within a single training set, unlike [1], which assumes incremental subpopulation learning.
>  - Methodology: Instead of using prototypes to balance learning vs. forgetting, we employ prototype-based classification with explicit diversification to enhance subgroup discovery.
> In response to the reviewer’s feedback, we will cite [1] and supporting references in the revised manuscript, highlighting how prototype-based methods and margin-enforce loss from boosting theory refine decision boundaries for subpopulation shift challenges, while clearly delineating our contributions. We hope this clarification strengthens the contextual positioning of our work.
>
> ### Relation to Broader Scientific Literature
> The reviewer highlights the relevance of our work to addressing the subpopulation shift problem. We would like to underscore the generality of the subpopulation shift phenomenon: as noted in [3], challenges such as class imbalance, attribute imbalance, and spurious correlations are all instances of the more general phenomenon of subpopulation shift. By targeting these core issues, our approach has broad applicability across domains like medical imaging, fairness in hiring, and large-scale recognition tasks, ultimately leading to more reliable, inclusive models.
>
> [3] Yang et al., 2023. Change is hard: a closer look at subpopulation shift
>
> ---
>
> **Thank you for taking the time to review our work. If our responses resolve your concerns, please kindly consider raising your score. Please feel free to reach out with any remaining questions.**

---

> > ### Comment · Reviewer_aEu5 · 2025-04-08
> >
> > Thank the author for the reply. I have read the author's rebuttal and other reviews and resolved my concerns. Overall, this is a good paper with clear motivation, and the methodology is sound and novel. Thus, I increase my score to Accept. Please incorporate the rebuttal and the discussion of the relevant literature in the final version. I look forward to future work in this direction.

---

> > > ### Author Response · Authors · 2025-04-09
> > >
> > > We thank the reviewer for the thoughtful feedback and for increasing the score. We’re glad to hear that the motivation and novelty of our work were well-received. We will incorporate the rebuttal clarifications and the discussion of relevant literature into the final version.

---

### Official Review · Reviewer_VrPG · 2025-03-14

**Overall Recommendation:** 3

**Summary:**

This paper tackles the problem of subpopulation shift in machine learning, where the proportions of different subgroups within a dataset change between training and testing. The authors propose a novel method called Diversified Prototypical Ensemble (DPE) to improve robustness to such shifts. DPE combines prototypical classifiers with an ensemble approach and explicit diversification strategies. The core idea is to train multiple prototype classifiers per class, encouraging them to learn different decision boundaries that capture various subpopulations within each class. Diversification is achieved through an inter-prototype similarity loss (IPS) and bootstrap aggregation. The main result is that DPE significantly improves worst-group accuracy (WGA) on real-world datasets, both with and without subgroup annotations during training.

**Claims And Evidence:**

The main claim is that DPE improves robustness to subpopulation shifts, as measured by WGA, compared to existing methods. The evidence provided is primarily empirical, based on experiments on several benchmark datasets. While the results show consistent improvements, the evidence is not entirely convincing due to concerns about the fairness of the comparisons (see below). The claims regarding the importance of diversification are supported by ablation studies.

**Essential References Not Discussed:**

The paper appears to cover the most essential references for the core problem and proposed method.

**Experimental Designs Or Analyses:**

The authors state that their initial ERM training stage ("ERM*") uses stronger data augmentation and longer training than the ERM implementation in Yang et al. (2023). They use published results for several baselines (ERM, CRT, ReWeightCRT, DFR from Yang et al., 2023; RWY and AFR from their original papers), but it seems not all of them use this stronger protocol. This introduces a confounder, making it unclear whether improvements are due to DPE itself or the stronger ERM training.

**Methods And Evaluation Criteria:**

The proposed method (DPE) is a reasonable approach for the problem. Combining prototypical networks with ensemble learning and explicit diversification is a novel and potentially effective strategy. The evaluation criteria, primarily worst-group accuracy (WGA), is appropriate for assessing robustness to subpopulation shifts. The choice of benchmark datasets (from Yang et al., 2023) is also standard in this area.

**Other Comments Or Suggestions:**

None.

**Other Strengths And Weaknesses:**

1. The method introduces several hyperparameters: the number of prototypes (N), the temperature ($\tau$), the IPS loss weight ($\alpha$), and the size of the subsets used for training. While the authors provide some details on how these were chosen, the sensitivity to these parameters (except N) isn't thoroughly explored.
2. The paper relies heavily on intuition and empirical results. There's no theoretical analysis explaining why the diversification strategies lead to improved worst-group accuracy, or if and why the prototypes align well with relevant subpopulations. A more formal understanding of the method's properties would be valuable.

**Questions For Authors:**

See above.

**Relation To Broader Scientific Literature:**

The paper builds upon several areas of related work, including subpopulation shift, prototypical networks, and ensemble learning.

**Theoretical Claims:**

The paper does not present any formal proofs or theoretical claims.

---

> ### Author Rebuttal · Authors · 2025-04-01
>
> Thank you for your detailed review. We appreciate your recognition of the novelty and relevance of DPE, its effective combination of prototypical networks and ensemble diversification, and its strong empirical performance on worst-group accuracy across standard benchmarks.
>
> Your main concerns include
> - (1) potential confounding due to stronger ERM* training compared to baseline implementations
> - (2) limited hyperparameter sensitivity analysis, particularly for τ, α, and subset size
> - (3) unclear prototype-subgroup alignment.
>
> We address each of these points below with new results, analyses, and clarifications. Full tables are provided at https://github.com/anonymous102030411/anon.
>
> ### 1. Ensuring a Fair Comparison with Prior Art
> We thank the reviewer for raising this important concern. To isolate the effect of our method (DPE) from stronger training protocols, we retrained all feature extractors using the exact setup from Yang et al. (2023)—matching architecture, augmentation, and training schedule. Applying DPE on top of these retrained backbones (see **[Table 2a](https://shorturl.at/bUCeK)**) confirms that DPE outperforms all reported baselines, showing the gains stem from DPE itself, not the ERM backbone.
>
> We also report results for ERM* + DPE, which includes stronger augmentation and longer training (see **[Table 2b](https://shorturl.at/3sAov)**). While these features boost performance, **DPE’s advantage remains consistent across both ERM and ERM*** **setups**.
>
> Regarding fairness in comparing ERM* + DPE to prior work, we note that several baselines use enhanced protocols:
> - DFR includes data augmentation.
> - RWG and RWY use extended training and thorough tuning.
> - CRT’s code includes augmentation, though not mentioned in the paper.
> - CnC, SSA, and GAP lack public code, so augmentation use is unclear.
>
> Thus, **comparing ERM*** **+ DPE to baseline reports is fair**, as many already benefit from stronger pipelines. In this context, DPE still achieves state-of-the-art worst-group accuracy across nearly all benchmarks.
> To support transparency, we will include:
> - A clear discussion of ERM vs. ERM* in the paper.
> - A full performance table with both ERM + DPE and ERM* + DPE results.
>
> ### 2. Hyperparameter Tuning and Sensitivity
> We thank the reviewer for pointing out the importance of hyperparameter sensitivity analysis. In our study, hyperparameters related to DPE—specifically the inverse temperature (**1/τ**), and the IPS loss weight (**α**)—are tuned using a held-out subset of the validation set, which is split into training and validation folds for tuning. Once optimal hyperparameters are selected, the prototypical ensemble is trained on the full validation set using these tuned values. Therefore, the subset size is not a hyperparameter in our pipeline.
> To address the reviewer’s concern, we conducted a sensitivity analysis on inverse temperature (**1/τ ∈ {10, 20, 30, 40}**) and IPS loss weight (**α ∈ {1e4, 5e4, 1e5, 5e5}**). Results across three datasets (Waterbirds, MetaShift, Living17) are presented in the attached tables (see **[Tables 7-10](https://shorturl.at/JZufO)**).
> The findings indicate that (1) DPE is robust to both τ and α on Waterbirds and MetaShift, with low standard deviations across tested values; (2) Living17 shows greater sensitivity, likely due to its more challenging subpopulation structure. Nonetheless, even in this case, worst-group accuracy varies within an acceptable range.
> **These results confirm that DPE’s performance is generally stable across a reasonable range of hyperparameter settings**. We’ll include this ablation study in the final version of the paper.
>
> ### 3. Prototype-subgroup alignment
> We appreciate the reviewer’s desire for deeper theoretical insight. To better understand whether learned prototypes capture meaningful subgroup structure, we conducted an exploratory qualitative analysis. Specifically, we tasked a third party (ChatGPT) with identifying common traits among the closest samples to each prototype—without providing group labels. This process revealed recurring semantic and ecological patterns (e.g., habitat type, pose, morphology) within prototype clusters, despite not being explicitly supervised to discover such groupings (see **[Figure 1 and 2](https://shorturl.at/o6tUI)**). These findings suggest that Diversified Prototypical Ensembles (DPE) may encourage meaningful prototype-subgroup alignment, potentially contributing to improved worst-group performance. We now clarify this insight in the revised draft.
>
> ---
>
> **Thank you again for your time and thoughtful review. If our responses addressed your concerns, we’d appreciate your consideration in raising the score. Please don’t hesitate to let us know if any points remain unclear—we’re happy to provide further clarification.**

---

> > ### Comment · Reviewer_VrPG · 2025-04-05
> >
> > The authors have addressed most of my concerns. Therefore I am raising my score to 3.

---

> > > ### Author Response · Authors · 2025-04-08
> > >
> > > Thanks for re-evaluating the submission. We’re pleased the changes resolved your concerns, and we’ll incorporate them into the final version.

---

### Official Review · Reviewer_rN4y · 2025-03-16

**Overall Recommendation:** 3

**Summary:**

This paper introduces the Diversified Prototypical Ensemble (DPE) to improve the robustness of machine learning models to subpopulation shifts. It replaces the standard linear classification layer with an ensemble of distance‐based prototypical classifiers. A two-stage training scheme is used: first use ERM to train a feature extractor and then fine-tune the ensemble on a held-out validation set. Empirical evaluations across nine diverse real-world datasets demonstrate that DPE is better in worst-group accuracy than state-of-the-art methods, both when subgroup annotations are available and unavailable.

**Claims And Evidence:**

The paper claimed, "These classifiers are trained using ... to maximize prototype diversity, ensuring robust subpopulation capture within each class". Although improved worst-group accuracy is provided, the evidence linking each learned prototype to a semantically meaningful subpopulation is somewhat ambiguous.

**Essential References Not Discussed:**

Not that I am aware of.

**Experimental Designs Or Analyses:**

The ablation study on the number of prototypes is somewhat insufficient. Why limit it to less than 15? I am curious what will happen if you further enlarge the number.

The authors claimed a weakness in their increased complexity, however, they did not conduct experiments to compare the running time, speed, computation resources, etc.

**Methods And Evaluation Criteria:**

For evaluation, the paper mainly focuses on worst group accuracy when compared with other methods. The performance of average accuracy is not completely reported.

**Other Comments Or Suggestions:**

NA

**Other Strengths And Weaknesses:**

One strength is the paper is well-organized and easy to follow. Some weaknesses include: no theoretical justification, not solid enough evidence to show why worst-group accuracy improved, and some missing ablation studies on important hyperparameters.

**Questions For Authors:**

NA.

**Relation To Broader Scientific Literature:**

The paper relates to work including prototypical networks, ensembling, and subpopulation shifts.

**Theoretical Claims:**

There are no theoretical claims or proofs.

---

> ### Author Rebuttal · Authors · 2025-04-01
>
> Thank you for your thoughtful review. We appreciate your recognition of our well-organized presentation, the strong empirical performance of DPE on worst-group accuracy, and its relevance to prototypical networks and subpopulation shift.
> You raised key concerns regarding:
>
> - (1) ablation on prototype count and hyperparameter sensitivity,
> - (2) missing runtime/resource analysis,
> - (3) incomplete reporting of average accuracy,
> - (4) limited evidence linking prototypes to meaningful subpopulations, and
> - (5) lack of theoretical justification.
>
> We address each of these below with new results, analyses, and clarifications. Full tables are provided at https://github.com/anonymous102030411/anon.
>
> ### 1. Further Ablation on the Number of Prototypes
>
> We thank the reviewer for raising this important question. To further investigate the effect of the number of prototypes per class, we extended our ablation study beyond 15 prototypes and now report results up to 40 prototypes per class.
> Let $\text{WGA}_N$ denote the worst-group accuracy (WGA) when using $N$ prototypes per class. We compute the percentage improvement over the single-prototype case as:
>
> $\Delta_N = \frac{\text{WGA}_N - \text{WGA}_1}{\text{WGA}_1} \times 100\%$
>
> We evaluated this on four representative datasets—Waterbirds, CelebA, Metashift, and CheXpert—under both settings: with and without subgroup annotations. The average percentage improvements are as follows:
>
> $\Delta_5 = 2.4\%$, $\Delta_{10} = 3.3\%$, $\Delta_{15} = 3.7\%$, $\Delta_{25} = 3.7\%$, $\Delta_{40} = 3.7\%$
>
> These results show that worst-group accuracy increases rapidly with the number of prototypes up to $N=15$, but plateaus beyond that point, indicating *diminishing returns*. Specifically, increasing from $N=15$ to $N=40$ provides no further gain in WGA.
>
> We interpret this as empirical evidence that a moderate number of diversified prototypes (e.g., 10–15 per class) is sufficient to capture the key subpopulation structures. Beyond this range, new prototypes tend to overlap in latent space with existing ones, limiting additional benefit. We also note that larger ensembles increase computational and memory costs without proportional gains. These observations justify our choice of using $N=15$ in the main experiments, balancing performance and efficiency.
> Regarding the performance sensitivity to other important hyperparameters in our study, we direct the reviewer to our response to Reviewer [VrPG](https://shorturl.at/t93so) with more extensive results and discussion.
>
> ### 2. Computational Complexity Analysis
>
> We clarify that “complexity” refers to implementation (e.g., added hyperparameters), not compute overhead. DPE adds minimal training cost, as prototypes act like lightweight linear layers. Compared to DFR, we train 15 simple classifiers instead of one. At inference, the overhead is negligible relative to the feature extractor.
>
> Table 1 (See **[Table 1](https://shorturl.at/KekLM)**) compares DPE and DFR in sample throughput and memory usage. Increasing from 15 to 100 prototypes only slightly raises time per batch (0.0031s→0.0045s) and keeps GPU memory below 1 GB—well within typical deep learning budgets. With fewer classes, the relative increase in complexity is even smaller. The complexity analysis will be included in the appendices of the revised version.
>
> ### 3. Reporting Average Accuracy
>
> We appreciate the suggestion to include average accuracy alongside worst-group accuracy. We revised the paper to include average accuracy in Table 6 (see **[Table 6](https://shorturl.at/KQEil)**), demonstrating that DPE remains competitive on average accuracy while prioritizing worst-group robustness.
>
> ### 4. Clarifying the Link Between Learned Prototypes and Subpopulations
>
> Our core contribution is demonstrating that prototype diversity reliably improves robustness across a wide range of subpopulation shift benchmarks. We provide the new qualitative visualization showing that learned prototypes naturally cluster semantically related samples and often align with meaningful subgroups, even without explicit supervision (see **[Figure 1 and 2](https://shorturl.at/o6tUI)**). This supports our central claim that diversity in prototype space enables better subgroup coverage. Expanded ablations further support the empirical foundation of our method.
>
> ### 5. Theoretical Justification
>
> The efficacy of DPE is supported primarily through extensive empirical evidence. We provide an intuitive rationale: by diversifying decision boundaries within each class, DPE encourages models to rely on more robust features rather than spurious correlations. We highlight this in the revised discussion and consider a formal theoretical study to be an important direction for future work.
>
> ---
>
> **Thank you for taking the time to review our work. If we have answered your questions, then we would appreciate you considering raising your score. If anything is still unclear, we are happy to clarify.**

---

> > ### Comment · Reviewer_rN4y · 2025-04-04
> >
> > My concerns are mostly addressed. Therefore I am glad to increase the score from 2 to 3.

---

> > > ### Author Response · Authors · 2025-04-08
> > >
> > > Thank you for taking the time to review our response. We're glad the revisions addressed your concerns and appreciate the updated score. Your comments helped clarify key points, and the changes will be included in the revised manuscript.

---

### Decision · Program_Chairs · 2025-05-01

**Decision:**

Accept (poster)

**Comment:**

The paper tackles the problem of subpopulation shifts by training a diverse ensemble of prototype classifiers on top of a pre-trained feature extractor. The core idea is to have each prototype focus on different aspects of the data, thereby discovering and correctly classifying diverse subpopulations without explicit annotations. The authors demonstrate this approach can achieve better worst-group accuracy compared to existing methods.

The review team generally found the work to be well-motivated, with a promising methodology that combines prototypical classifiers with ensemble learning and explicit diversification strategies. The extensive empirical evaluations across nine diverse real-world datasets were highlighted as a strength. One key concern revolved around the lack of grounded justification or intuition for the proposed method and why the diversification strategies lead to improved worst-group accuracy. While I won’t repeat all dimensions of improvement called out in the review process, I urge the authors to work on additional ablation studies on the number of prototypes, HP sensitivity, and the potential confounding effect of the stronger ERM method.  As a general practice, include average accuracy alongside worst-group accuracy will provide additional context for the readers.